# PML restrains p53 activity and cellular senescence in clear cell renal cell carcinoma

Matilde Simoni [1], Chiara Menegazzi [1], Cristina Fracassi [1], Claudia C Biffi[1,8], Francesca Genova [2], Nazario Pio Tenace [3], Roberta Lucianò[3], Andrea Raimondi[4], Carlo Tacchetti [4,5], James Brugarolas [6,7], Davide Mazza[4] & Rosa Bernardi [1]✉

## Abstract

**Clear-cell renal cell carcinoma (ccRCC), the major subtype of RCC, is frequently diagnosed at late/metastatic stage with 13% 5-year disease-free survival. Functional inactivation of the wild-type p53 protein is implicated in ccRCC therapy resistance, but the detailed mechanisms of p53 malfunction are still poorly characterized. Thus, a better understanding of the mechanisms of disease progression and therapy resistance is required. Here, we report a novel ccRCC dependence on the promyelocytic leukemia (PML) protein. We show that *PML* is overexpressed in ccRCC and that PML depletion inhibits cell proliferation and relieves pathologic features of anaplastic disease in vivo. Mechanistically, PML loss unleashed p53-dependent cellular senescence thus depicting a novel regulatory axis to limit p53 activity and senescence in ccRCC. Treatment with the FDA-approved PML inhibitor arsenic trioxide induced PML degradation and p53 accumulation and inhibited ccRCC expansion in vitro and in vivo. Therefore, by defining non-oncogene addiction to the PML gene, our work uncovers a novel ccRCC vulnerability and lays the foundation for repurposing an available pharmacological intervention to restore p53 function and chemosensitivity.**

**Keywords** PML; ccRCC; p53; Senescence; Arsenic Trioxide
**Subject Categories** Cancer; Urogenital System

## Introduction

Oncogenesis is a complex developmental process that relies on genetic alterations of cancer genes as well as non-genetic activation of supportive gene networks that provide essential contributions to disease pathogenesis, a phenomenon known as "non-oncogene addiction" (Nagel et al, 2016). Most of these activated non-oncogenes operate within stress-responsive pathways and grant adaptation to cellular perturbations triggered by oncogenic signals. The promyelocytic leukemia (*PML*) gene is a general sensor of cellular stresses that include viral infections, oxidative stress, and DNA damage and promotes a variety of adaptive cell-protective responses (Lallemand-Breitenbach and de Thé, 2018). Mechanistically, PML moieties nucleate biomolecular condensates mostly located in the nucleus and named PML nuclear bodies (PML-NBs) that act as structural platforms for the assembly and regulation of protein complexes (Lallemand-Breitenbach and de Thé, 2018). A long list of PML interactors has been compiled based on studies performed in different cell types and physio-pathological conditions, leading to the current model of a highly adaptable protein with versatile and tissue-specific functions (Hsu and Kao, 2018). This is particularly evident in cancer, where PML is at large nongenetically deregulated and exerts oncogenic or tumor-suppressive functions depending on the tumor context (Datta et al, 2020). As examples, PML opposes tumorigenesis in acute promyelocytic leukemia (APL), the disease where it was originally identified (de Thé et al, 2017), and solid tumors like prostate and lung cancer (Chen et al, 2018; Wang et al, 2017), but holds oncogenic activities in triple-negative breast cancer (TNBC) (Carracedo et al, 2012; Martín-Martín et al, 2016; Ponente et al, 2017; Arreal et al, 2020), ovarian cancer (Gentric et al, 2019; Liu et al, 2017), and glioma (Amodeo et al, 2017; Aldaz et al, 2022; Kuwayama et al, 2009; Iwanami et al, 2013; Tampakaki et al, 2021). Renal cell carcinoma (RCC) is a tumor type where PML functions are still poorly characterized.

RCC accounts for ~2% of cancer diagnoses and deaths and it comprises different histological and molecular subtypes (Pontes et al, 2022). The most common subtype is clear-cell RCC (ccRCC), an aggressive disease that is often diagnosed at late/metastatic stages where the 5-year disease-free survival drops to 13% as opposed to 70–90% for localized tumors (SEER, National Cancer Institute 2021). Therefore, an improved understanding of ccRCC pathological mechanisms is necessary to identify novel vulnerabilities and improve patients' outcome.

In a study investigating the role of the SCP1 phosphatase, which promotes PML degradation, it was suggested that PML exerts tumor-suppressive functions in ccRCC (Lin et al, 2014). However,

[1]Division of Experimental Oncology, IRCCS San Raffaele Scientific Institute, Milan, Italy. [2]Center for Omics Sciences, IRCCS San Raffaele Scientific Institute, Milan, Italy. [3]Department of Pathology, IRCCS San Raffaele Scientific Institute, Milan, Italy. [4]Experimental Imaging Center, IRCCS San Raffaele Scientific Institute, Milan, Italy. [5]Universita' Vita-Salute San Raffaele, Milan, Italy. [6]Kidney Cancer Program, Simmons Comprehensive Cancer Center, University of Texas Southwestern Medical Center, Dallas, TX, USA. [7]Department of Internal Medicine, Division of Hematology/Oncology, University of Texas Southwestern Medical Center, Dallas, TX, USA. [8]Present address: Medical Advisor, Sanofi, Milan, Italy. ✉E-mail: bernardi.rosa@hsr.it

more recent studies reported that *PML* is overexpressed in ccRCC and belongs to gene signatures that identify high-risk patients and correlate with worse disease outcome (Wu et al, 2023; Luo et al, 2023; Wang et al, 2023; Li et al, 2019), implying oncogenic activities. Beside this correlative evidence however, a rigorous functional assessment of the involvement of PML in this disease is still lacking.

Aiming to fill this gap, we investigated the function of PML in ccRCC via genetic and pharmacologic approaches and found that PML is essential for ccRCC expansion and its inhibition unleashes p53-dependent cellular senescence. Our findings describe for the first time a pathological tumor setting where PML opposes p53 activation. Because p53 is rarely mutated in ccRCC but is functionally inactivated via poorly characterized molecular mechanisms (Amendolare et al, 2022), our findings identify a novel clinically actionable gene network that may be targeted to ameliorate ccRCC response to therapies.

## Results

### The PML protein is overexpressed in ccRCC

It was recently reported that *PML* is overexpressed in ccRCC compared to normal kidneys and belongs to a high-risk gene signature associated with worse outcome (Wu et al, 2023; Luo et al, 2023; Wang et al, 2023; Li et al, 2019). To determine if the PML protein is overexpressed cell-autonomously within ccRCC cells, we compared PML expression in ccRCC cell lines (Caki-1, RCC4, A49786-O and the patient-derived xenograft cell line UTSW-XP258, hereafter referred to as XP258) (Elias et al, 2021) with breast cancer (MDA-MB-231, MDA-MB-468, MCF7) and glioblastoma (U87) cell lines, as representative of tumor types where PML is reportedly overexpressed (Carracedo et al, 2012; Martín-Martín et al, 2016; Ponente et al, 2017; Amodeo et al, 2017; Aldaz et al, 2022; Kuwayama et al, 2009; Iwanami et al, 2013; Tampakaki et al, 2021). As a negative control, prostate cancer PC3 cells represented a tumor type where PML is downregulated (Chen et al, 2018).

Because PML moieties shuttle from a soluble nucleoplasmic distribution to insoluble PML-NBs where the PML protein is SUMOylated, whole cell lysates were prepared in Laemmli buffer to facilitate the detection of both soluble and insoluble PML species (Bercier et al, 2023). By this analysis, we identified various PML isoforms, which derive from well-described alternative splicing of the *PML* gene (Uggè et al, 2022), along with high molecular weight PML-SUMO conjugates (Bercier et al, 2023) (Fig. 1A). Overall, expression of PML isoforms and SUMOylated PML species was higher in ccRCC cells than all other cell lines analyzed (Fig. 1A). Mass spectrometry pan-cancer proteomic data provided by the NCI Clinical Proteomic Tumor Analysis Consortium (CPTAC) (Clark et al, 2019) confirmed that the PML protein is markedly overexpressed in ccRCC (kidney renal clear-cell carcinoma; KIRC) compared to paired normal adjacent tissue (Fig. 1B). PML overexpression was confirmed also in glioblastoma multiforme (GBM) and breast cancer (BRCA), albeit being more evident in ccRCC (Fig. 1B). Of note, this analysis showed similar levels of PML protein across ccRCC, breast cancer and glioblastoma multiforme specimens (Fig. 1B), unlike cell lines, where PML

upregulation was more evident in ccRCC (Fig. 1A). This may depend on different contributions by the tumor micoreonvironment or other technical aspects related to proteins extraction from tumor samples. Analysis of Cancer Cell Line Encyclopedia (CCLE) mRNA expression data confirmed that PML levels are highest in RCC, followed by head and neck squamous carcinoma (HNSC) and glioma (Fig. 1C). These data were further validated in The Cancer Genome Atlas (TCGA) specimens, where *PML* is overexpressed in KIRC, as previously observed (Wang et al, 2023), as well as in HNSC and GBM (Fig. 1D). Of note, this analysis showed *PML* overexpression also in papillary renal cell carcinoma (kidney renal papillary carcinoma; KIRP), albeit to a lesser extent (Fig. 1D). Finally, *PML* overexpression identified patients with decreased survival in TCGA-KIRC, as previously observed (Wang et al, 2023), as well as in TCGA-GBM, but not in TCGA-KIRP and TCGA-HNSC (Fig. EV1A–D).

Collectively, these results extend previously published observations by confirming that PML is overexpressed at the mRNA and protein level in ccRCC specimens, and showing that PML upregulation is tumor cell-intrinsic. Because *PML* is rarely mutated in ccRCC (0.2%, source: cBioPortal), PML overexpression is unlikely to be due to genetic alterations. In addition, high PML expression correlated with decreased survival in clear cell but not in papillary renal cancer (Fig. EV1A,B), suggesting that it may exert oncogenic functions specifically in this tumor subtype.

To understand if PML overexpression resulted in increased formation of PML-NBs in ccRCC, we compared cell lines representative of ccRCC and TNBC and observed higher numbers of PML-NBs/cell in ccRCC cell lines (Fig. EV1E). Moreover, in analyzing these data, we noticed that ccRCC nuclei had less diffuse PML signal when compared to TNBC nuclei (Fig. 1E), suggesting that higher numbers of PML-NBs may be accompanied by increased partitioning of PML into PML-NBs.

The biogenesis of PML-NBs is a multi-step process that begins with oxidation and oligomerization of nucleoplasmic PML moieties, followed by recruitment of the SUMO-conjugating enzyme UBC9, PML SUMOylation and PML-NBs maturation (Abou-ghali and Lallemand-Breitenbach, 2024). To understand if ccRCC cells had increased biogenesis of PML-NBs, we quantitatively compared nuclear PML distribution in RCC4 and MDA-MB-231 cells as representative of ccRCC and TNBC, respectively. Segmentation and quantification of nuclei and PML-NBs revealed that RCC4 cells have larger nuclei than MDA-MB-231 cells and display higher PML intensity and more PML-NBs per nucleus as well as per unit area, indicating that they have higher PML-NBs content that is not solely due to larger nuclear area (Fig. EV1F–I). To quantify PML aggregation into PML-NBs, we measured the intensity of PML immunofluorescence signal within PML-NBs vs a 2-pixel wide nucleoplasmic ring around PML-NBs. The ratio of PML intensity inside/outside PML-NBs was higher in RCC4 than in MDA-MB-231 cells (Fig. 1E), indicating that PML is enriched in PML-NBs in RCC4 cells.

To validate these data with a biochemical approach, cells were lysed either in RIPA buffer, which does not extract highly insoluble NB-bound PML, or in Laemmli buffer. By performing densitometric analysis of the largest PML isoform (PML-I; *ca* 110 kDa) together with its SUMOylated species (higher bands contained within the dashed line; Fig. 1F), we observed that the normalized amount of Laemmli-extracted PML is higher in RCC4 than in

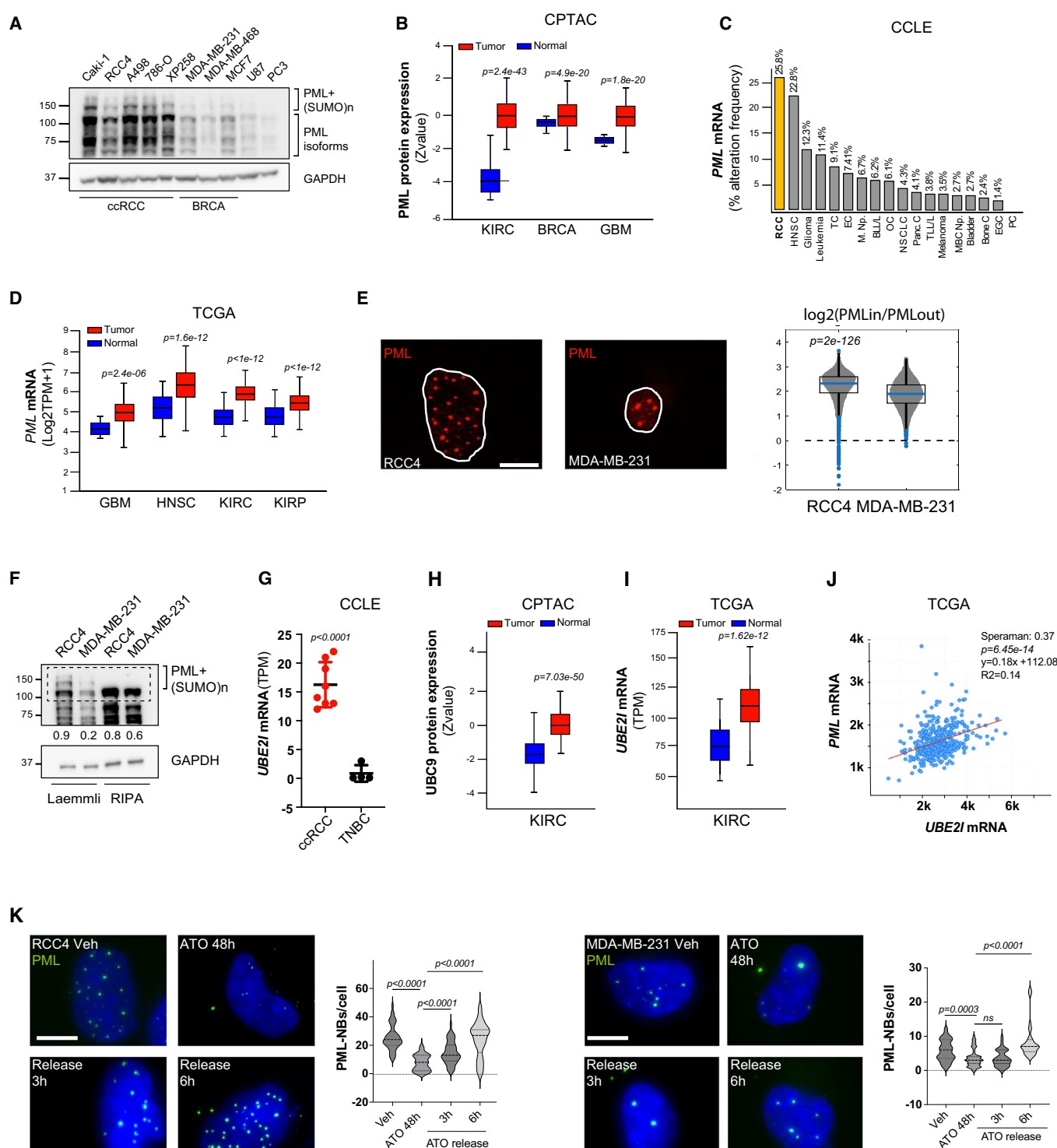

MDA-MB-231 cells (0.9 vs 0.2; Fig. 1F), while RIPA-extracted PML-I is similar in the two cell lines (0.8 vs 0.6; Fig. 1F). This analysis further indicates that PML is more concentrated within insoluble PML-NBs in RCC4 cells.

Of note, analysis of UBC9 expression (*UBE2I* gene) in cell lines from CCLE revealed higher UBC9 levels in ccRCC than in TNBC cells (Fig. 1G), suggesting that in ccRCC PML may accumulate

within the PML-NBs due to overexpression of its SUMO-conjugating enzyme. In line with this hypothesis, we observed UBC9 mRNA and protein overexpression in ccRCC tumor samples (Fig. 1H,I), and positive correlation of UBC9 and PML mRNA expression (Fig. 1J).

Thus, to understand if increased UBC9 expression is accompanied by enhanced PML-NBs biogenesis in ccRCC cells, we

◄ **Figure 1. PML is overexpressed in ccRCC and efficiently assembles into PML-NBs.**

(A) Immunoblot blot analysis showing PML expression in the indicated renal cell carcinoma (RCC), breast cancer (BRCA), glioblastoma and prostate cancer cell lines. Several PML isoforms and high molecular weight PML SUMOylated species are detected. Numbers represent densitometric analysis of PML levels normalized over GAPDH. Molecular weight markers (kDa) are shown on the left. The blot represents one out of three independent experiments with similar results. (B) PML protein abundance expressed as Z-value in tumor and matched normal tissues in the Clinical Proteomic Tumor Analysis Consortium (CPTAC) datasets: kidney renal clear-cell carcinoma (KIRC, $n = 169$ normal adjacent and $n = 219$ tumor samples), breast cancer (BRCA, $n = 18$ normal adjacent and $n = 125$ tumor samples), and glioblastoma multiforme (GBM, $n = 10$ normal adjacent and $n = 99$ tumor samples). Data are represented in box and whisker plots where the central band denotes the median value, box contains interquartile ranges, while whiskers mark minimum and maximum values. (Source: UALCAN). (C) Ranking of cell lines of the cancer cell line encyclopedia (CCLE) based on the frequency of cells with high PML mRNA expression within each group. Renal cell carcinoma (RCC $n = 31$, in bold), head and neck squamous cancer (HNSC, $n = 35$), glioma ($n = 93$), leukemia ($n = 48$), thyroid cancer (TC, $n = 11$), endometrial cancer (EC, $n = 27$), myeloproliferative neoplasms (M. Np., $n = 15$), B-lymphoblastic leukemia/lymphoma (BLL/L, $n = 17$), ovarian cancer (OC, $n = 64$), non-small cell lung cancer (NSCLC, $n = 163$), pancreatic cancer (Panc. C, $n = 57$), T-lymphoblastic leukemia/lymphoma (TLL/L, $n = 37$), melanoma ($n = 106$), mature B-cell neoplasia. (MBC, Np $n = 112$), bladder ($n = 44$), bone cancer (Bone C, $n = 89$), esophagogastric cancer (EGC, $n = 84$), prostate cancer (PC, $n = 8$) (Source: cBioPortal). (D) Expression levels of *PML* mRNA in tumors and normal tissues in the indicated TGCA (The Cancer Genome Atlas) datasets: GBM ($n = 5$ normal and $n = 156$ tumor samples); HNSC ($n = 44$ normal and $n = 520$ tumor samples); KIRC ($n = 72$ normal and $n = 533$ tumor samples); KIRP ($n = 32$ normal and $n = 290$ tumor samples). Data are represented in box and whisker plots where the central band denotes the median value, box contains interquartile ranges, while whiskers mark minimum and maximum values. (Source: UALCAN). (E) Representative images of PML immunofluorescence in RCC4 and MDA-MB-231 cells (left). A single z-stack is shown. Scale bar 10 μm. PML is shown in red and white contours delineate nuclei. Quantification data showing the distribution of PML in PML-NBs and in a 2-pixel wide nucleoplasmic ring around PML-NBs (right; RCC4, $n = 124$; MDA-MB-231, $n = 150$). Data are expressed as $\log_2(PML_{IN}/PML_{OUT})$ and are represented in box and whisker plots where the central band denotes the median value, box contains interquartile ranges, while whiskers mark the minimum and maximum values. Individual data points are shown. Statistical significance was calculated with the Bonferroni corrected Kolmogorov–Smirnov test. (F) Immunoblot blot analysis showing PML protein levels in RCC4 and MDA-MB-231 cells upon protein extraction with Laemmli or RIPA buffers. Several PML isoforms and high molecular weight PML SUMOylated species are detected. The dashed line contains PML-I and PML SUMOylated species. Numbers represent densitometric analysis of PML species contained within the dashed line and normalized over GAPDH. Molecular weight markers (kDa) are shown on the left. The blot represents one out of three independent experiments with similar results. (G) UBC9 mRNA abundance in ccRCC ($n = 8$) and TNBC ($n = 4$) cell lines extracted from CCLE. Data represent mean ± SD (Student's *t* test). (Source: CCLE). (H) UBC9 protein abundance expressed as Z-value in tumor and matched normal tissues in the CPTAC-KIRC dataset (KIRC, $n = 169$ normal adjacent and $n = 219$ tumor samples). Data are represented in box and whisker plots where the central band denotes the median value, box contains interquartile ranges, while whiskers mark minimum and maximum values. (Source: UALCAN). (I) Expression levels of *UBE2I* (UBC9) mRNA in tumors and normal tissues in the TGCA-KIRC dataset ($n = 72$ normal and $n = 533$ tumor samples). Data are represented in box and whisker plots where the central band denotes the median value, box contains interquartile ranges, while whiskers mark minimum and maximum values. (Source: UALCAN). (J) Spearman's correlation analysis of *PML* mRNA and *UBE2I* (UBC9) mRNA abundance in the TCGA-KIRC dataset ($n = 72$ normal and $n = 533$ tumor samples) (Source: cBioPortal). (K) Representative images of PML immunofluorescence in RCC4 and MDA-MB-231 cells after ATO treatment and release for the indicated hours (h; left). Scale bars 10 μm and 20 μm, respectively. Quantification data of PML-NBs per cell (right; $n = 46$ RCC4 cells/condition; $n = 45$ MDA-MB-231 cells/condition). PML is shown in green and nuclei were counterstained with DAPI (blue). Data represent mean ± SD of three independent experiments (Student's *t* test). Source data are available online for this figure.

compared the recovery time of PML-NBs formation upon PML degradation by arsenic trioxide (ATO). By binding PML B-box-2 domain, ATO leads to PML nuclear aggregation followed by proteasomal degradation (Bercier et al, 2023). 48 h treatment of RCC4 and MDA-MB-231 cells with ATO led to a significant reduction of PML-NBs in both cell lines (Fig. 1K). Of note, ATO also induced the accumulation of aberrant cytoplasmic PML aggregates that reportedly built upon ATO due to the blockade of post-mitotic PML nuclear recycling (Lång et al, 2012). Release from ATO led to a faster recovery of PML-NBs in RCC4 than in MDA-MB-231 cells (3 h vs 6 h; Fig. 1K).

Taken together, these data show that in ccRCC cells PML overexpression is accompanied by enhanced PML-NBs formation and suggest that this may be prompted by UBC9 overexpression.

## PML sustains proliferation and tumor expansion in ccRCC

To investigate the function of PML in ccRCC, we first attempted to deplete PML by constitutive expression of PML-specific shRNAs but were unable to expand ccRCC cell lines with PML knockdown, suggesting that ccRCC cells may depend on PML for in vitro expansion. To test this hypothesis, we used a doxycycline-inducible knockdown system. PML downregulation upon induction of two independent PML shRNAs in Caki-1, RCC4, and A498 cells significantly impaired cell proliferation and focus-forming capacity compared to cells transduced with scramble shRNA (Fig. 2A–D). PML depletion exerted more modest effects of growth inhibition in

TNBC cells MDA-MB-231 and papillary RCC cells ACHN (Fig. EV2A–F), indicating that ccRCC cells are more sensitive to PML depletion.

The proliferation arrest induced by PML knockdown was accompanied by cell accumulation in G0/G1 (Figs. 2E and EV3A,B) and decreased BrdU incorporation (Figs. 2F and EV3C), without signs of apoptosis (Fig. EV3D). Similar results were obtained in patient-derived xenograft (PDX) cells XP258 (Fig. EV3E–J). In summary, these data show that ccRCC cells depend on PML expression for proliferation.

To validate our findings in vivo, we selected Caki-1 and A498 cells that reportedly engraft in immunocompromised mice, while RCC4 cells do not consistently form tumors in vivo (Brodaczewska et al, 2016). Doxycycline administration was initiated when Caki-1 and A498 tumors reached 200 mm³ (Fig. 3A,B). PML knockdown, which was verified by IHC at the experimental endpoint (Fig. 3C,D), dramatically reduced tumor expansion (Fig. 3A,B). Immunohistochemical analysis of the proliferation marker KI67 showed that PML silencing impaired cancer cell proliferation in both models (Fig. 3E,F), although data in A498 tumors were not statistically significant (Fig. 3F), possibly because control tumors showed vast necrotic areas and reduced KI67 positivity at the experimental endpoint.

Histopathological evaluation revealed that tumors generated with both cell lines were of high grade (mostly nucleolar grade III–IV), with reduced tumor grade upon PML knockdown (control tumors: 7/7 grade IV; PML-depleted tumors: 4/7 grade III, 3/7 grade IV). Moreover, while control tumors (7/7) displayed

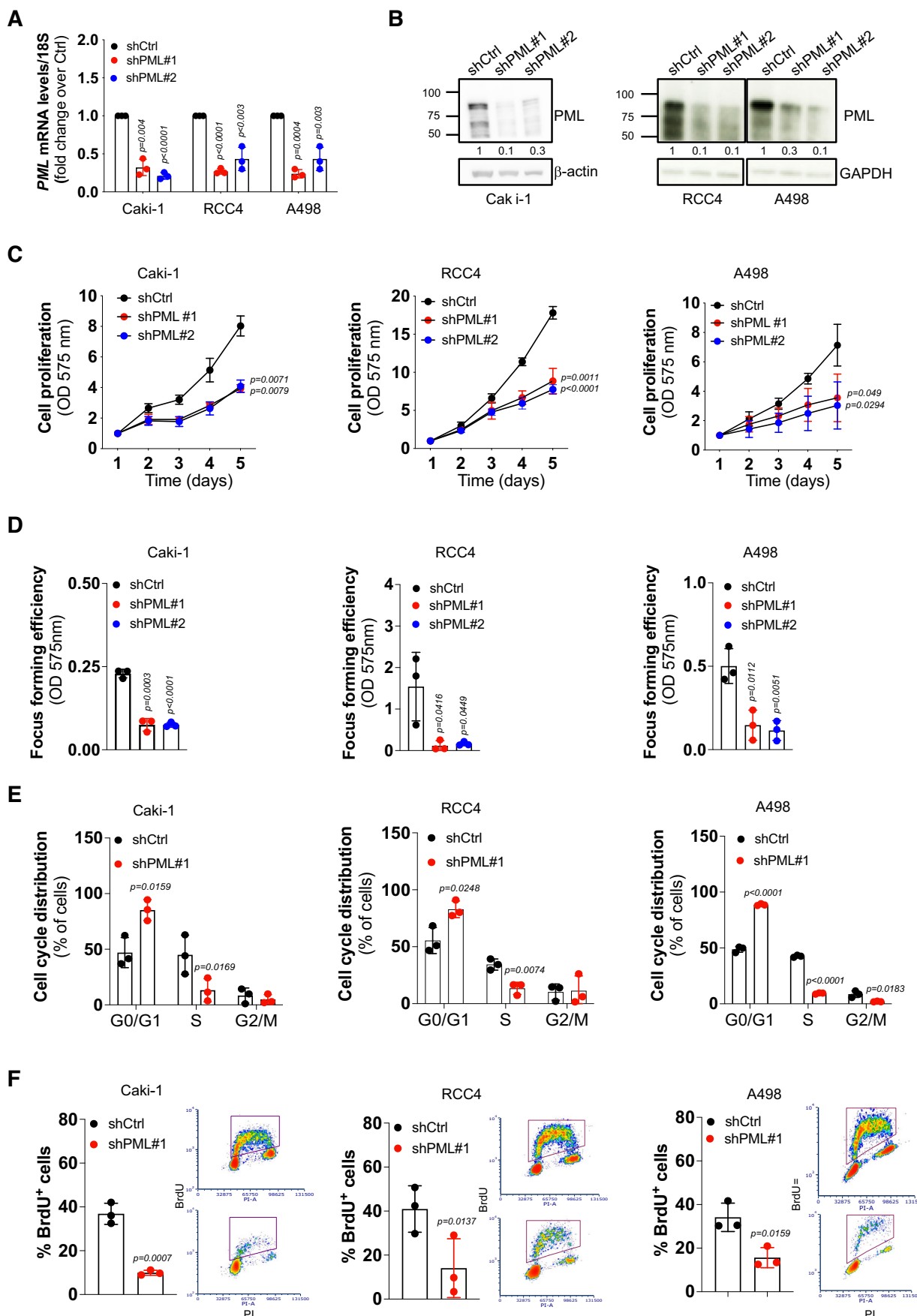

**Figure 2.  PML knockdown inhibits ccRCC proliferation and cell cycle progression.**

(A) qPCR analysis showing the silencing efficiency of two independent shRNAs against PML (shPML#1 and shPML#2) or a scrambled shRNA sequence (shCtrl) in the indicated cell lines. Data represent mean ± SD of three biological replicates (Student's *t* test). (B) Immunoblot analysis showing silencing efficiency of two independent shRNAs against PML (shPML#1 and shPML#2) or a scrambled shRNA sequence (shCtrl) in the indicated cell lines. β-actin or GAPDH were used as loading control. Numbers represent densitometric analysis of PML levels normalized over β-actin or GAPDH. Molecular weight markers (kDa) are shown on the left. Blots represent one out of three independent experiments with similar results. (C–E) Proliferation curves (C), focus-forming assays (D) and cell cycle distribution by FACS analysis (E) of the indicated cell lines expressing shPML#1, shPML#2 or shCtrl. (C) Data are shown as fold change of OD 575 nm measurements over day 1. Data represent mean ± SD of three biological replicates (Student's *t* test). (F) Evaluation of BrdU-positive cells by FACS analysis in the indicated ccRCC cell lines expressing shPML#1 or shCtrl (left). Scatter plot of FACS analysis of BrdU-positive cells (right). Data represent mean ± SD of three biological replicates (Student's *t* test). Scatter plots show one out of three independent experiments with similar results. Source data are available online for this figure.

anaplastic features typical of aggressive ccRCC (Elias et al, 2021), including large areas of sarcomatoid or rhabdoid differentiation (Fig. 3G), PML-depleted tumors conserved a distinctive eosinophilic clear-cell morphology (7/7) and showed signs of remission in some cases (2/7 were small masses characterized by few tumor cells embedded in fibrotic tissue; Fig. 3G).

Taken together, these results demonstrate that ccRCC cells are addicted to PML expression, thus suggesting a novel non-oncogenic dependency in this tumor type.

## PML regulates a pro-proliferative transcriptional program in ccRCC

To get mechanistic insights into PML functions in ccRCC, we profiled the PML-regulated transcriptome in RCC4 cells 96 h after induction of PML silencing (Fig. 2A,B). We found that PML regulates the expression of 2192 genes, with 936 genes being upregulated and 1256 genes downregulated upon PML knockdown (Dataset EV1). Functional enrichment analysis of genes downregulated upon PML silencing identified gene families linked to cell cycle progression and proliferation (Fig. 4A), in line with the phenotype of PML-depleted cells (Fig. 2). Included in these gene sets are cell cycle regulators like *CDK1, CDK2, Cyclin E2, POLA1, POLE, E2Fs*, and *FOXM1* (Dataset EV1).

Interestingly, genes suppressed by PML (Dataset EV1) fell into more heterogeneous gene families associated to four major biological processes: senescence, extracellular matrix organization, inflammation/interferon signaling, and cholesterol biosynthesis (Fig. 4B). PML was reported to regulate all of these processes, albeit in different cellular systems and sometimes with opposite outcomes. For example, PML inhibits cholesterol biosynthetic pathways also in prostate cancer (Chen et al, 2018), but in apparent contrast with our observations PML promotes the expression of extracellular matrix (ECM) remodeling factors in TNBC (Fracassi et al, 2023), stimulates inflammation/interferon signaling in different cell types (Hsu and Kao, 2018), and induces p53-dependent senescence upon oncogenic insults (Hsu and Kao, 2018). Nevertheless, the upregulation of p53-dependent and senescence gene networks (Fig. 4B) is consistent with the cell cycle arrest induced by PML knockdown in ccRCC cells (Figs. 2 and EV3). In addition, the removal of PML leads to senescence induction also in TNBC and glioblastoma (Arreal et al, 2020; Aldaz et al, 2022). Along these lines, it is now appreciated that the secretome of senescent cells (senescence-associated secretory phenotype, or SASP) is enriched in proteins that promote ECM remodeling, like metalloproteases, along with inflammatory cytokines and chemokines (Faget et al, 2019).

To understand whether gene families linked to ECM remodeling and inflammation upregulated in RCC4 cells with low PML levels belong to a SASP, we overlapped all genes negatively regulated by PML with a PML-dependent senescence secretome defined in TNBC cells (Arreal et al, 2020) or a generic SASPome profiled in fibroblasts subjected to various senescence inducers (Basisty et al, 2019). We found that 15 PML-regulated genes in ccRCC overlapped with the PML-regulated secretome of TNBC cells (Arreal et al, 2020) with an enrichment of gene families linked to ECM organization (Fig. 4C). Also, many genes upregulated upon PML silencing in RCC4 cells belong to conserved fibroblasts SASPomes and pertain to both ECM organization and inflammation (Fig. 4D).

In conclusion, our data suggest that the products of these genes may belong to a SASP, thus reducing the complexity of the upregulated transcriptome of PML-silenced cells to one of senescence.

Taken together, our transcriptomic analysis confirms that PML is essential for ccRCC proliferation and reveals that its inhibition elicits a transcriptional response that is strongly suggestive of cellular senescence.

## PML inhibition unleashes cellular senescence

Senescence is characterized by a reshaping of cell morphology and intracellular organization, with cell flattening and enlargement, accumulation of cytoplasmic granular material, and loss of nuclear integrity (Herranz and Gil, 2018). Accordingly, we found that PML knockdown in RCC4, Caki-1, and A498 cells led to increased cell size (FSC-A) and granularity (SSC-A) (Fig. 5A), accompanied by cell flattening (Fig. EV4A) and depletion of the structural component of the nuclear envelope lamin B1 (Fig. 5B). Microscopic observation of PML-depleted cells revealed that enlarged, senescent-like cells were characterized by vacuolation and accumulation of cytoplasmic granular material (Fig. EV4A). Because lysosomes have been proposed to be the major component of cytoplasmic aggregates in senescent cells (Herranz and Gil, 2018), we measured the area occupied by lysosomes in the cytoplasm of PML-depleted cells. Immunofluorescence staining of the LAMP2 lysosomal marker showed increased lysosome occupancy in RCC4 and A498 cells upon PML knockdown (Fig. 5C), which was confirmed by transcriptional upregulation of lysosomal gene families (Fig. 4B) and ultrastructural cell morphology examination revealing accumulation of degradative structures akin to lysosomes (Fig. EV4B). The most established biomarker of senescence is the lysosomal protein β-galactosidase (Basisty et al, 2019). Surprisingly, we did not detect senescence-associated β-galactosidase (SA-β-gal) positive cells 5 days upon induction of PML silencing (Appendix Fig. S1). However, SA-β-gal accumulation was observed at later times of PML silencing (Fig. 5D). Moreover, continuous cell

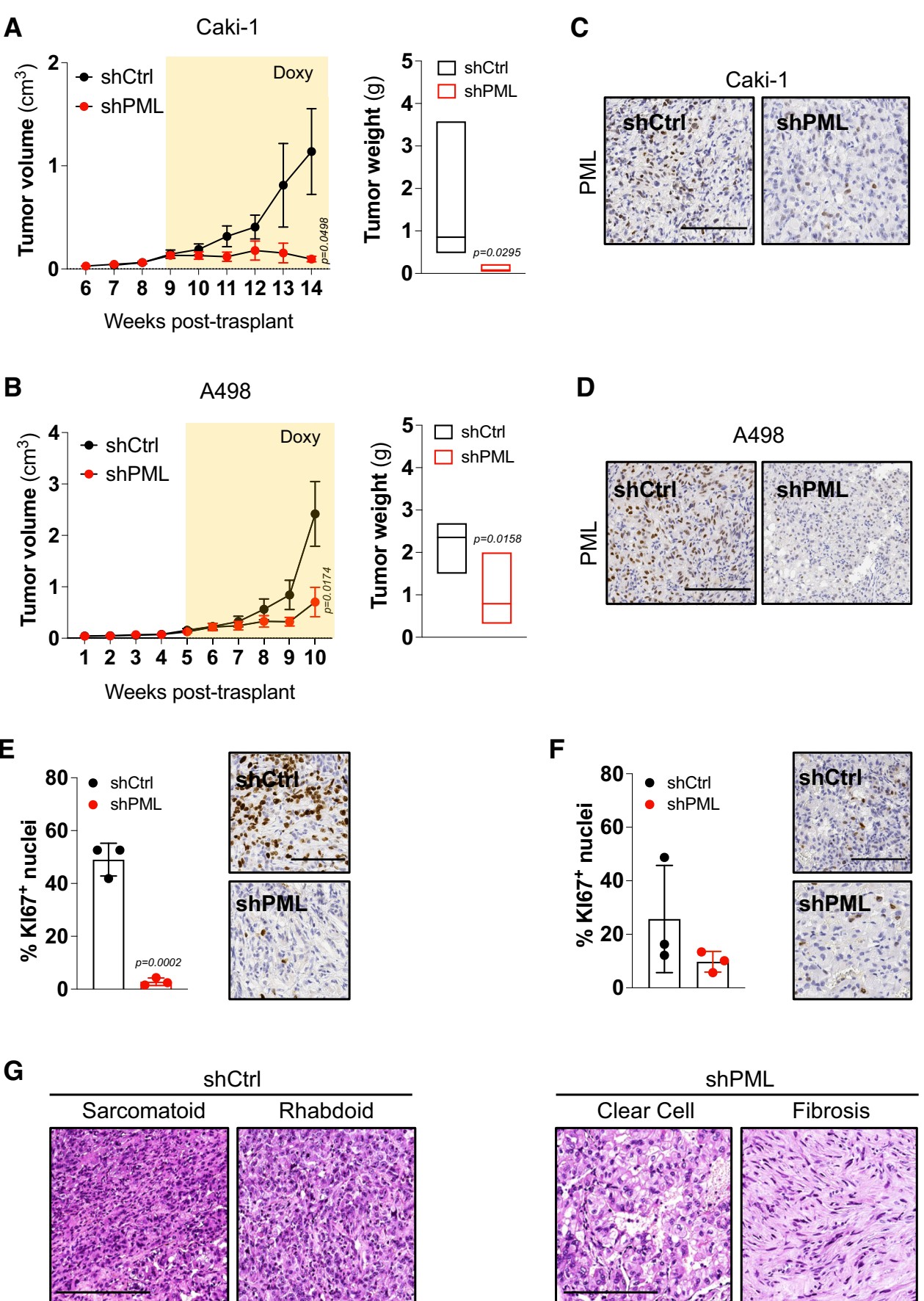

◀ **Figure 3. PML sustains ccRCC expansion.**

(A, B) Tumor progression of Caki-1 (A) and A498 (B) cells expressing shCtrl ($n = 6$) or shPML ($n = 6$) injected subcutaneously in immunocompromised mice (left). Tumor volumes were measured at indicated days upon injection. Yellow boxes represent time periods of doxycycline treatment. Data represent mean ± SEM (Student's $t$ test). Tumor weights at experimental endpoints indicated in (A, B) are shown in the right panels. Data are presented in box plots where the central band denotes the median value and containing interquartile ranges. (C, D) Representative images of immunohistochemical PML analysis in tumor tissues at the experimental endpoint in Caki-1 (C) and A498 (D) xenografts. Scale bar 100 μm. (E, F) Representative images of KI67 immunohistochemical analysis in tumor tissues at the experimental endpoint in Caki-1 (E) and A498 (F) xenografts (right). Scale bar 100 μm. Quantification data (left) represent mean ± SD of three biological replicates (Student's $t$ test). (G) Representative images of hematoxylin and eosin staining showing sarcomatoid and rhabdoid anaplastic features, or eosinophilic clear-cell morphology and tissue fibrosis in shCtrl and shPML tumors, as indicated. Scale bar 100 μm. Source data are available online for this figure.

replating in a 3T3 protocol revealed that the proliferation arrest induced by PML targeting is stable (Fig. 5E). Taken together, these data show that PML depletion elicits senescence in ccRCC cells.

## PML inactivation induces a p53-dependent growth arrest

Although PML was long described as a negative regulator of p53 (Lallemand-Breitenbach and de Thé, 2018), the upregulation of p53 transcriptional networks upon PML silencing (Fig. 4B) led us to investigate the involvement of p53 downstream PML depletion. This is a relevant question in the context of ccRCC, because p53 is rarely mutated but is maintained in an inactive state via poorly characterized mechanisms in this tumor, and its stabilization leads to cellular senescence (Amendolare et al, 2022; Xie et al, 2021).

PML knockdown led to the accumulation of p53 and its bona fide transcriptional target p21 in all ccRCC cell lines analyzed (Fig. 6A). Of note, analysis of DNA damage foci marked by phosphorylated γ-H2AX did not reveal increased DNA damage upon PML silencing (Appendix Fig. S2), suggesting that p53 induction is DNA damage-independent. In line with our observations, induction of senescence upon inhibition of oncogenic signals like c-Myc was also reported to occur via a p53-dependent and DNA damage-independent mechanism (Wu et al, 2007).

To assess whether PML depletion-induced growth arrest depends on p53, we overexpressed a dominant negative p53 mutant (p53DD) that inhibits wild-type p53 activity by abrogating DNA binging (Gottlieb, 1998). p53DD or empty vector (EV) were transduced in RCC4 and A498 cells carrying doxycycline-inducible control or PML-specific shRNAs (Fig. 6B) and p53DD expression rescued the proliferation defects caused by PML depletion (Fig. 6C). To corroborate these findings, we took advantage of 786-O ccRCC cells carrying p53 mutations (c.560-2 A > G and c.832 C > G) (Leroy et al, 2014). PML silencing failed to inhibit cell proliferation and focus-forming capacity in 786-O cells (Fig. 6D–F), as opposed to the other ccRCC cell lines (Fig. 2), indicating that ccRCC cells with mutant p53 are not affected by PML depletion.

In sum, our results suggest that, unlike the senescence process that is induced by PML depletion in TNBC, which is p53-independent (Arreal et al, 2020), suppressing PML expression in ccRCC provokes a phenotype of growth arrest that is mediated by p53 and its target gene network. Thus, a therapeutic approach that targets PML could serve as a novel strategy to reactivate wild-type p53 and induce tumor regression.

## Arsenic trioxide blocks ccRCC expansion

To further demonstrate ccRCC addiction to PML expression, we exploited PML targetability with ATO in ccRCC models, including

cell lines and a PDX-derived cellular model (Elias et al, 2021; Chen et al, 2016). Forty-eight hours of treatment of ccRCC cells with the standard dose of 1 μM ATO and lower concentrations (0.25 and 0.5 μM) revealed sizable PML degradation in all cell lines (Fig. 7A–D, right panels). Importantly, ATO treatment inhibited cell proliferation (Fig. 7A–D, left panels) and focus-forming capacity (Fig. EV5A) in a dose-dependent manner. Taken together, the sensitivity of ccRCC cell lines to ATO was comparable to NB4 cells, which are representative of acute promyelocytic leukemia where ATO is part of standard-of-care therapy (Lo-Coco et al, 2016), and higher than TNBC MDA-MB-231 cells, where ATO reportedly inhibits PML oncogenic functions (Martín-Martín et al, 2016) (Fig. EV5B,C). Of note, ATO treatment was overall more efficient than PML silencing at inducing PML depletion and proliferation arrest in vitro (Figs. 7A–D and 2B,C) and also led to substantial apoptosis in all cell lines analyzed (Fig. EV5D). These data suggest that severe PML depletion, which can be achieved by using ATO at concentrations within the therapeutic range, may tilt the senescence phenotype toward cell death. Induction of apoptosis in most ccRCC cell lines was similar to that induced by ATO in NB4 cells, while MDA-MB-231 cells were more resistant to ATO-induced cell death (Fig. EV5E). Finally, the reduction of ccRCC expansion upon ATO treatment was confirmed in vivo with A498-derived tumors (Fig. 7E).

Similar to PML silencing (Fig. 6A), treatment with ATO led to p53 upregulation in ccRCC cells (Fig. 7F), suggesting that this compound could be used to unleash the tumor-suppressive functions of wild-type p53 in this tumor type. In line with this hypothesis, inhibition of endogenous p53 via p53DD expression in Caki-1, RCC4, and A498 cells led to significant recovery of cell expansion upon ATO treatment (Fig. EV5G). Rescue of ATO-induced growth suppression by p53DD was less evident at higher ATO concentrations (Fig. EV5G), which may be caused by accumulating oxidative stress and/or p53-independent mechanisms.

Interestingly, exposure to ATO triggered growth suppression and apoptosis also in 786-O cells (Figs. 7G and EV5F), which carry p53 mutations and did not respond to PML silencing (Fig. 6D–F). However, it was recently reported that ATO can reactivate p53 mutants that carry structural variations within the DNA binding domain via direct binding and allosteric activation, leading to the rescue of p53 transcriptional activity and tumor-suppressive functions (Chen et al, 2021). Although p53 mutations in 786-O cells are poorly characterized (Chen et al, 2021), one of these variants (P278A) maps close to a structural p53 mutant that was shown to be reactivated by ATO (R282W) (Chen et al, 2021). To test the dependency of ATO efficacy on mutant p53 expression in 786-O cells, we knocked down endogenous p53 by shRNA

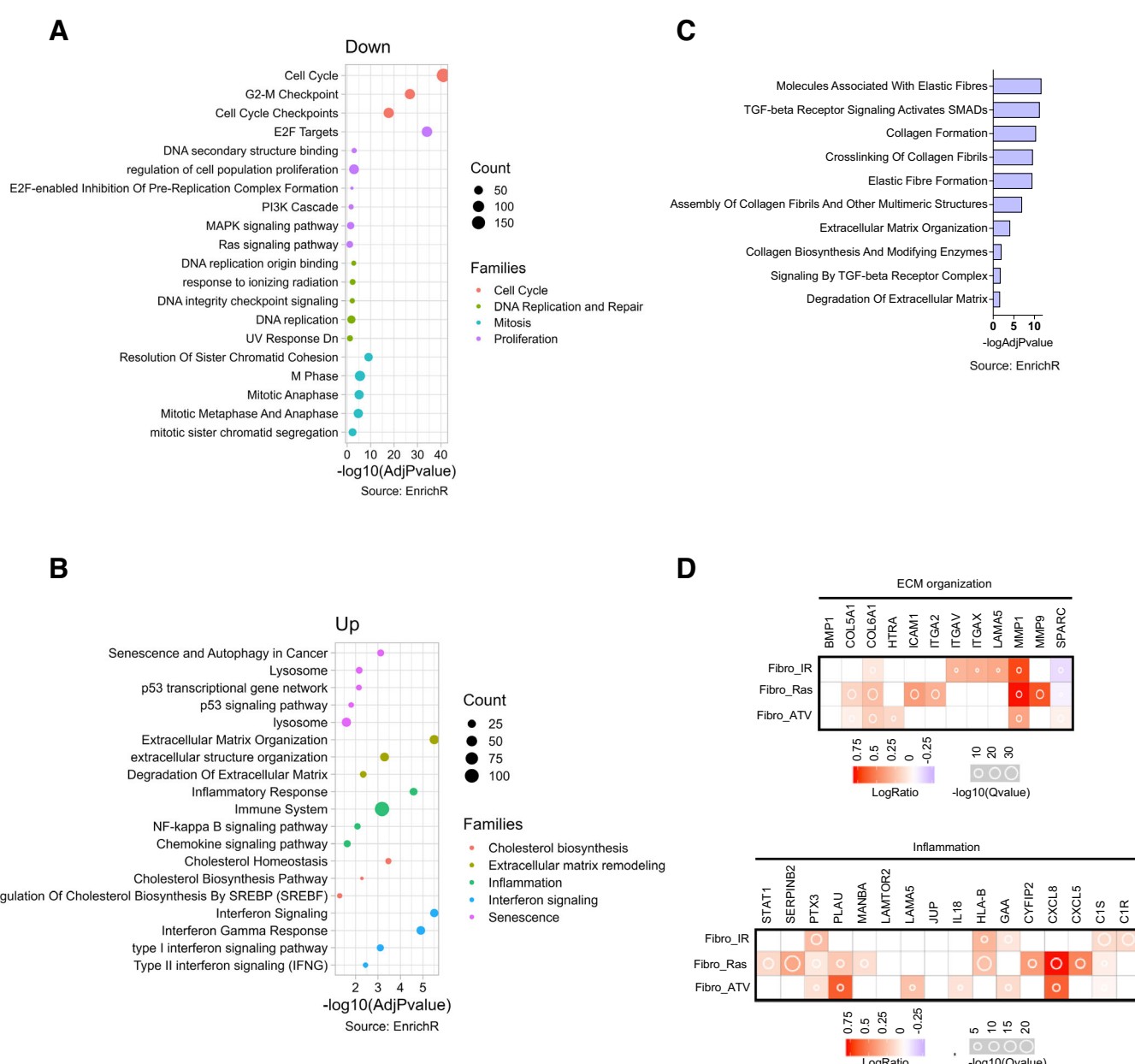

**Figure 4. PML silencing leads to transcriptional regulation of proliferation and senescence gene families.**

(A, B) Gene set enrichment analysis of differentially expressed genes (significant threshold of 0.05, adjusted *P* value by False Discovery Rate) downregulated (A) and upregulated (B) in RCC4 cells upon PML silencing (96 h). The Gene Count value represents the number of genes regulated by PML contained in each gene set. Functional categories were grouped in color-coded superfamilies based on similarity of function. Dot sizes represent the number of genes in each term. The results from Fisher's exact tests are represented as −log(adjusted *P* value). (C) Functional enrichment analysis of genes upregulated upon PML silencing (96 h) in RCC4 and overlapping with SASP factors regulated by PML in MDA-MB-231 cell lines. Results are represented as −log(adjusted *P* value). (D) The abundance of SASP factors produced by fibroblasts undergoing senescence via irradiation (IR), RasV12 overexpression (Ras) or Atazanavir (ATV) and overlapping with genes involved in ECM organization and inflammation upregulated upon PML silencing in RCC4 (96 h). Results are represented as −log(adjusted *Q*-value) (Source: SASP Atlas).

(Fig. EV5H). p53 knockdown induced a slight but significant proliferation defect in 786-O cells (Fig. EV5H), suggesting that at least one of the p53 mutations reported in this cell line acts as a gain-of-function. Notably, silencing of p53 reduced the proliferation blockade triggered by ATO in 786-O cells (Fig. 7H), indicating that ATO-induced growth suppression is mediated at least in part by p53 in these cells. This was confirmed by expressing the p53DD

protein, which also partially rescued ATO-induced growth arrest (Fig. EV5I). As observed in ccRCC cell lines carrying wild-type p53, the rescue of ATO-induced growth suppression was less evident at higher ATO concentrations or upon prolonged exposure (Figs. 7H and EV5I).

Collectively, these results show that ccRCC cells are sensitive to ATO and suggest that the tumor-inhibitory effects of this

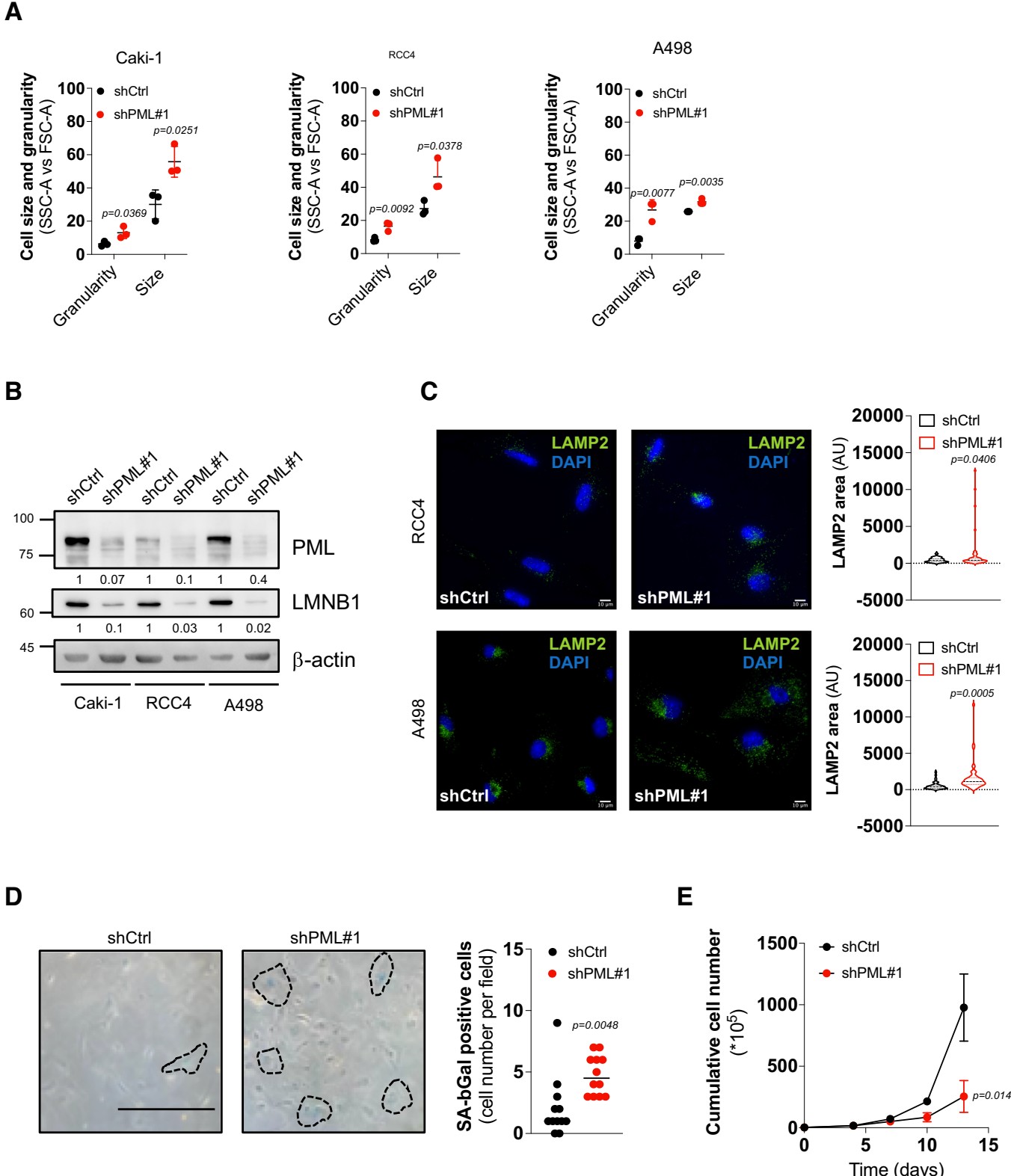

◄ **Figure 5. PML knockdown triggers cellular senescence.**

(A) Percentage of cells characterized by increased size (FSC-A) and granularity (SSC-A) analyzed by FACS in the indicated cell lines expressing shPML#1 or shCtrl. Data represent mean ± SD of three biological replicates (Student's *t* test). (B) Immunoblot of PML and LMNB1 in the indicated cells expressing shPML#1 or shCtrl. β-actin was used as a loading control. Numbers represent densitometric analysis of PML and LMNB1 levels normalized over β-actin. Molecular weight markers (kDa) are shown on the left. The blot represents one out of three independent experiments with similar results. (C) Representative images of intracellular LAMP2 distribution by immunofluorescence (green) in the indicated cell lines expressing shPML#1 or shCtrl (left). Quantification data of LAMP2 area occupancy (RCC4, *n* = 53 cells per group, A498, *n* = 43 cells per group) are shown on the right. Nuclei were counterstained with DAPI (blue). Data represent mean ± SD of three independent experiments (Student's *t* test). (D) Representative phase contrast images of SA-β-gal positive cells in RCC4 cells expressing shCtrl or shPML#1 (left). Scale bar 100 μm. Quantification data of the average number of SA-β-gal positive cells/fields in 12 fields (right). Data represent mean ± SD of three independent experiments (Student's *t* test). (E) Cumulative cell number of RCC4 cells expressing shPML#1 or shCtrl propagated with a 3T3 protocol. Data represent mean ± SD of three independent experiments (Student's *t* test). Source data are available online for this figure.

compound can be exerted at least in part via pharmacological targeting of PML in p53 proficient cells as well as reactivation of p53 mutants, thus setting the basis for ATO repurposing in ccRCC.

## Discussion

In this study, we reveal a previously unknown non-oncogene addiction to the *PML* gene in ccRCC by finding that PML is overexpressed in this tumor and that its genetic or pharmacologic inhibition triggers tumor-suppressive responses of cellular senescence and p53 induction.

PML was initially described as a protein endowed with tumor-suppressive functions in cancer, but accumulating evidence points to a more complex scenario of tumor suppression or promotion that depends on the tissue context (Datta et al, 2020). So far, the function of PML in kidney cancer has been poorly investigated, with few studies providing evidence of a tumor-suppressive role (Lin et al, 2014; Bernardi et al, 2011). Specifically, in a genetic model of papillary renal cancer triggered by *Tsc2* heterozygous inactivation, *Pml* genetic deletion accelerated tumor progression, and this was accompanied by increased mTOR activity in the kidney (Bernardi et al, 2011). However, this work focused on papillary kidney cancer and did not address the function of PML in ccRCC. This was investigated in a later study, where PML was positioned downstream the SCP1 phosphatase, a protein with tumor-suppressive functions in ccRCC that promotes PML stabilization (Lin et al, 2014). Although this paper clearly defined the function of SCP1 in ccRCC, the role of PML was not addressed unambiguously, due to a lack of functional assays where PML inactivation was investigated in the absence of SCP1 manipulation (Lin et al, 2014). In this work, it was also observed that PML is downregulated at the protein level in ccRCC specimens compared to matched normal tissue (Lin et al, 2014). In contrast with these findings, recent studies found that *PML* is overexpressed in ccRCC and belongs to prognostic high-risk signatures (Li et al, 2019; Luo et al, 2023), thus pointing to an oncogenic function. Our work confirms that PML is upregulated in ccRCC (at the mRNA and protein level) in tissues and cell lines, thus implying that previously reported evidence of PML downregulation (Lin et al, 2014) may be due to patients' selection or use of a specific antibody for immunohistochemical studies. Our results are based on publicly available mass spectrometry datasets, thus avoiding possible biases due to antibody specificity/selectivity.

The molecular causes of PML overexpression in ccRCC are currently unknown. The *PML* gene is regulated in a highly tissue-specific manner in cancer. For example, it is downregulated in many tumor types like lung and prostate cancer (Wang et al, 2017; Chen et al, 2018) but is overexpressed in other tumors including TNBC and glioma (Carracedo et al, 2012; Martín-Martín et al, 2016; Amodeo et al, 2017). In the latter, transcriptional upregulation of *PML* was positioned downstream the oncogenic transcription factors STAT3, HIF-1α, and SOX9 (Martín-Martín et al, 2016; Ponente et al, 2017; Aldaz et al, 2022). In addition, PML expression and its aggregation into PML-NBs was linked to increased oxidative phosphorylation and reactive oxygen species in a subtype of ovarian cancer (Gentric et al, 2019). Taken together, these studies place PML within transcriptional networks of stress responses and stem cell maintenance, in line with the role of PML in these biological processes (Hsu and Kao, 2018; Vogiatzoglou et al, 2022). Albeit currently unknown, PML upregulation may depend on stress-responsive pathways also in ccRCC. For instance, activation of hypoxia signaling is a pathogenetic hallmark of ccRCC (Meléndez-Rodríguez et al, 2018) and PML is a HIF-1α target gene in TNBC (Ponente et al, 2017).

Interestingly, in addition to showing that PML is upregulated in ccRCC, we also find that PML SUMOylation and aggregation into PML-NBs is increased compared to TNBC cells, where PML is also overexpressed with respect to normal tissue. This is accompanied by UBC9 overexpression in ccRCC cell lines and patients' samples. Thus, the converging effects of PML upregulation and enhanced assembly of PML-NBs may prompt a heightened oncogenic activity of PML in this tumor type. Accordingly, ccRCC cell lines with constitutive PML depletion could not be established, and inducible PML silencing led to a profound and prolonged state of cell cycle arrest in vitro and in vivo. Nonetheless, surprisingly, PML is not scored as a ccRCC dependency gene in Cancer Dependency Map (DepMap, source: https://depmap.org), a genome-wide CRISPR and shRNA screen designed to identify essential genes across hundreds of cancer cell lines. The reasons for this apparent discrepancy are presently unknown. However, because efforts like DepMap were developed to assay gene dependencies at a population level based on loss of cell viability, and because PML inhibition induces a state of prolonged cell cycle arrest, we speculate that maintenance of live cells with PML depletion may underestimate its contribution to ccRCC fitness in DepMap.

PML was reported to sustain tumor progression via many mechanisms that include upholding cancer stem cell maintenance (Carracedo et al, 2012; Ito et al, 2008; Amodeo et al, 2017), impinging on metabolic pathways (Carracedo et al, 2012) and promoting metastasis (Ponente et al, 2017). Mechanistically, these manyfold functions are achieved by regulating the expression and/

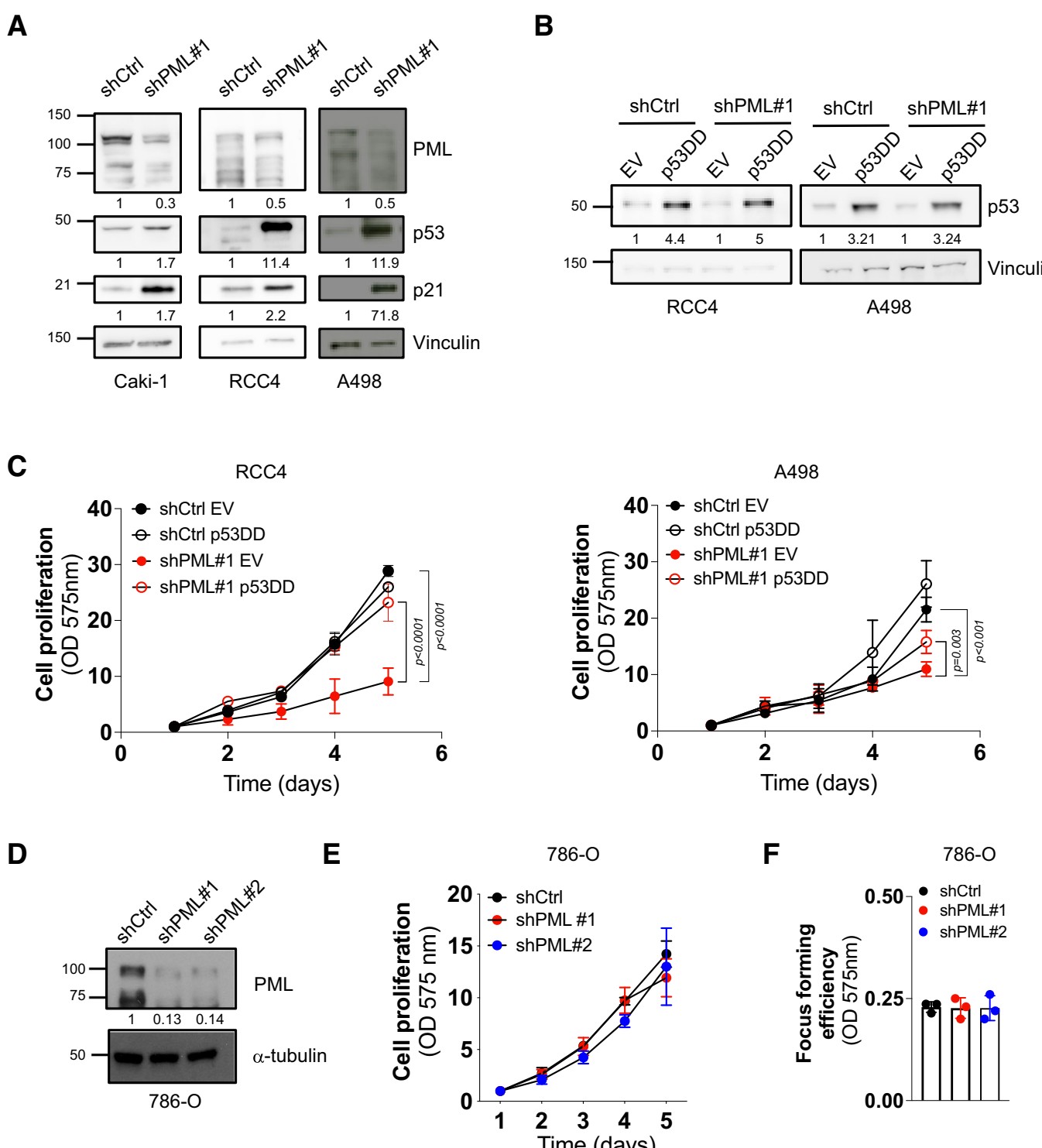

or activity or critical oncogenes and stem cell factors like mTOR, Sox9, Stat3, HIF-1α and c-Myc (Carracedo et al, 2012; Martín-Martín et al, 2016; Ponente et al, 2017; Ito et al, 2008; Arreal et al, 2020). Of note, in no previous cases it was reported that PML opposes p53 functions to favor tumor progression. Conversely, ample literature has described PML as a promoter of p53 activity, especially p53-induced senescence, albeit primarily in untrans-formed cells (Liebl and Hofmann, 2022). Along these lines, although recent data has provided convincing evidence that PML inhibits senescence in contexts where it performs oncogenic functions, like TNBC, this occurred via p53-independent mechanisms (Arreal et al, 2020).

◄
**Figure 6. Growth arrest induced by PML silencing is p53-dependent.**

(A) Immunoblot of PML, p53 and p21 in the indicated cell lines expressing shPML#1 or shCtrl. Vinculin was used as loading control. Numbers represent densitometric analysis of PML, p53 and p21 levels normalized over β-actin. Molecular weight markers (kDa) are shown on the left. The blot represents one out of three independent experiments with similar results. (B) Immunoblot analysis showing p53 expression in the indicated cell lines expressing inducible shPML#1 or shCtrl and transduced with p53DD or an empty vector (EV). Vinculin was used as loading control. Numbers represent densitometric analysis of p53 levels normalized over β-actin. Molecular weight markers (kDa) are shown on the left. The blot represents one out of three independent experiments with similar results. (C) Proliferation assays in RCC4 and A498 cells expressing shPML#1 or shCtrl and transduced with p53DD or EV. Data are shown as fold change of OD 575 nm measurements over day 1. Data represent mean ± SD of three independent experiments (Student's t test). (D) Immunoblot analysis showing silencing efficiency of two independent shRNAs against PML (shPML#1 and shPML#2) or a scrambled shRNA sequence (shCtrl) in the 786-O cell line. α-tubulin was used as loading control. Numbers represent densitometric analysis of PML levels normalized over α-tubulin. Molecular weight markers (kDa) are shown on the left. The blot represents one out of three independent experiments with similar results. (E, F) Proliferation (E) and focus-forming (F) assays of 786-O cell line expressing shPML#1, shPML#2 or shCtrl. (E) Data are shown as fold change of OD 575 nm measurements over day 1. Data represent mean ± SD of three biological replicates (Student's t test). Source data are available online for this figure.

In this scenario, a relevant finding of our work is that in ccRCC PML acts as a negative regulator of p53. Although this finding may appear in contrast with published literature on PML and p53 (Liebl and Hofmann, 2022), it reflects a phenomenon that was previously described for bona fide oncogenes like c-Myc, namely that p53-dependent stress responses are triggered both by oncogene overexpression in normal cells (oncogene-induced senescence or OIS) (Zindy et al, 1998), as well as by oncogene removal in fully transformed cancer cells (oncogene inhibition-induced senescence OIIS) (Wu et al, 2007). Albeit the mechanism of p53 induction upon PML removal in ccRCC remains to be clarified, our findings reveal that the relationship between PML and p53 is more complex and context-dependent than anticipated. Interestingly, ccRCC is a tumor type where the *TP53* gene is infrequently mutated yet the p53 protein is kept in check via poorly defined molecular mechanisms (Amendolare et al, 2022). Thus, our work involves the participation of PML to p53 regulation in ccRCC providing a previously unknown avenue to exploit p53 reactivation, one that can be pharmacologically explored with an FDA-approved leukemia treatment that promotes PML degradation: arsenic trioxide. Of note, we find that arsenic trioxide has a wider efficacy than PML blockade in ccRCC, as it effectively inhibits ccRCC expansion also in cells carrying p53 mutations. Our data support recent findings that described the direct binding and reactivation of structural p53 mutants by arsenic trioxide toward p53-induced tumor suppression (Chen et al, 2021). Thus, our work prompts future investigations into repurposing arsenic trioxide as a novel ccRCC therapeutic strategy.

## Methods

### Cell culture

Human RCC cell lines ACHN [the American Type Culture Collection (ATCC) number: CRL-1611] and RCC4 [European Collection of Authenticated Cell Lines (EACC) number 03112702], and TNBC cells MDA-MB-231 (ATCC number: HTB-26) and MDA-MB-468 (ATCC number: HTB-132) were cultured in DMEM (Euroclone ECM0103L). ccRCC cells Caki-1 (ATCC number: HTB-46), 786-O (ATCC number: CRL-1932), prostate cancer cells PC3 (ATCC number: CRL-1435) and APL cells NB4 (ATCC number: HB12215) were maintained in RPMI 1640 (Euroclone ECB2000). ccRCC A498 cells (ATCC number: HTB-44) were maintained in

MEM (Life Technologies 11095-080) supplemented with 1% MEM Non-Essential Amino Acids (Life Technologies 11140050) and 1% pyruvate. Cell lines were used at early passages (1–20) and were not authenticated. XP258 PDX-derived cells (Chen et al, 2016) were maintained in MEM (Life Technologies 11095-080) supplemented with 1% MEM Non-Essential Amino Acids (Life Technologies 11140050), 10 ng/ml EGF (Life Technology 13247-051), and 0.4 μm/ml Hydrocortisone (Merck H0396). For lentiviral production, HEK293T (ATCC number: CRL-11268) cells were grown in IMDM. For retroviral production, Phoenix-AMPHO (ATCC number: CRL-3213) cells were maintained in DMEM (ECB1072L). All culture media were supplemented with heat-inactivated 10% FBS (Euroclone ECS5000L) and 1% P/S (Euroclone ECB 3001). All cells were tested periodically for *Mycoplasma* negativity and maintained in a humidified atmosphere containing 5% $CO_2$ and 20% $O_2$.

### Reagents and treatments

Doxycycline monohydrate was purchased from Merck (D1822-500MG) and added to cell culture media at a final concentration of 100 ng/ml for 96 h to achieve PML silencing. Culture media was changed every 48 h. For in vivo experiments, doxycycline hyclate was purchased from Merck (D9891-1G). ATO used for in vitro and in vivo studies was purchased from Merck (A1010) and used at the indicated time points and concentrations. Nutlin-3 was purchased by Merck (N6287) and used at 10 nM for 6 days.

### Animal models

Animal studies were approved by the San Raffaele Institutional Animal Care and Use Committee (IACUC, protocol number 989). Mice were purchased from Charles River Laboratories, maintained in a pathogen-free animal facility, and treated in accordance with European Union guidelines. Specifically, mice were housed in transparent and individually ventilated cages (5 mice/cage maximum) with a minimum enclosure size of 330 cm²/mouse. Food and water were provided in one end of the cage leaving the other end free for urination and defecation. The bedding was provided in wood shavings and was changed weekly. All animals utilized in this study were male. For in vivo tumor growth assay, $4 \times 10^6$ A498 and $10 \times 10^6$ Caki-1 cells expressing doxycycline-inducible shCtrl or shPML (shPML#1) were injected subcutaneously in the right flank of 6–8 weeks old male NSG mice. Tumor volumes were

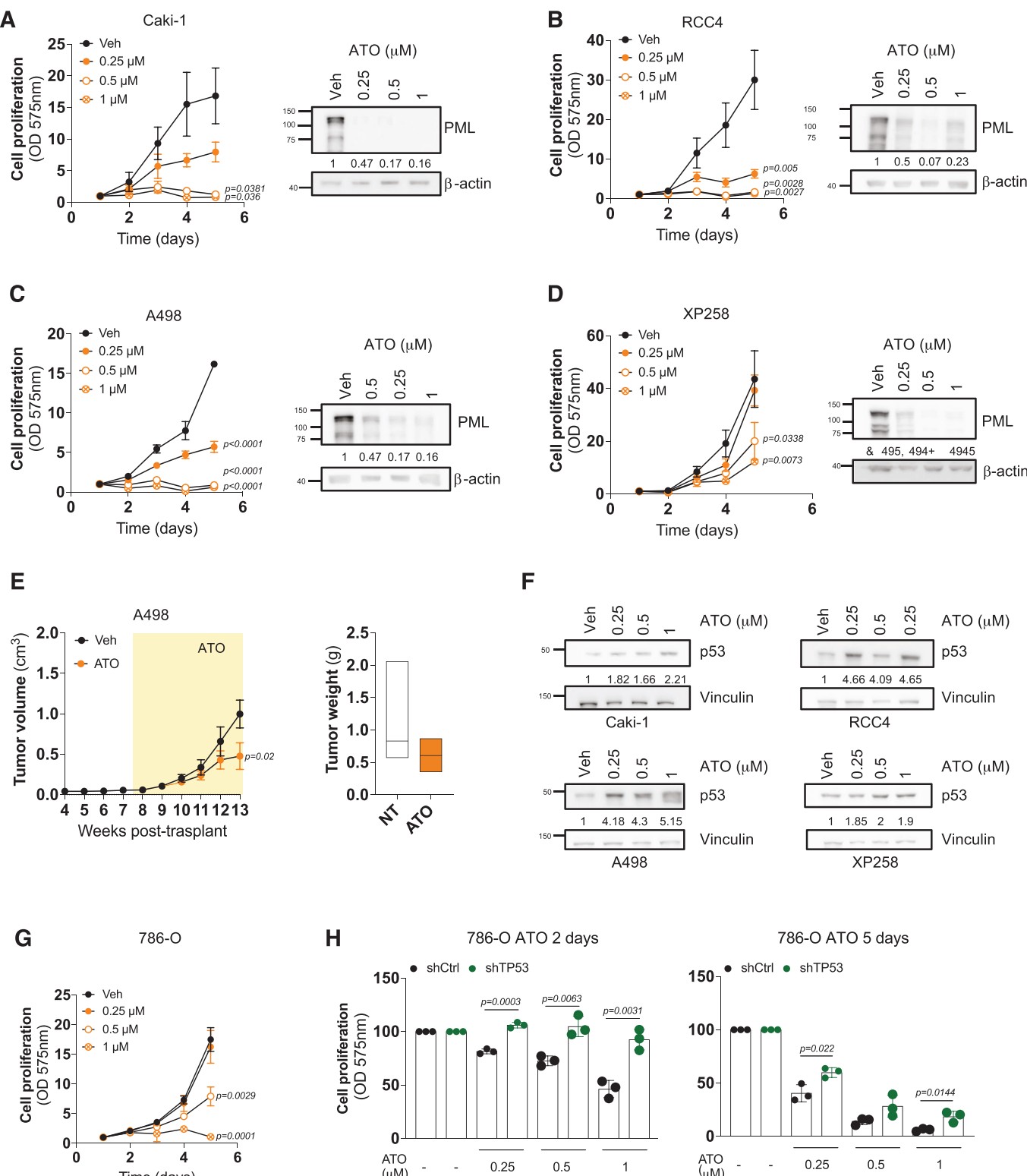

measured weekly using a caliper. When tumor masses reached 0.2 cm³, doxycycline hyclate was administered daily via oral gavage (25 mg/Kg) until the indicated experimental endpoint. For in vivo experiments with ATO, $4 \times 10^6$ A498 or $10 \times 10^6$ Caki-1 cells were injected subcutaneously into the right flank of NSG mice. When tumor masses reached 0.2 cm³, mice were randomly distributed in two cohorts and treated intraperitoneally with vehicle or ATO (4 mg/Kg). Tumor growth was measured weekly using a caliper and

◄ **Figure 7.  Arsenic trioxide blocks ccRCC expansion and induces p53 accumulation and apoptosis.**

(A–D) Proliferation assays of the indicated cell lines upon ATO treatment (0.25–1 μM) for 5 days (left). Growth curves are shown as fold change of OD 575 nm measurements over day 1. Data represent mean ± SD of three independent experiments (Student's t test). Immunoblot analyses of PML expression upon 48 h of ATO treatment (0.25–1 μM) in the indicated cell lines are shown on the right. Numbers represent densitometric analysis of PML levels normalized over β-actin. Molecular weight markers (kDa) are shown on the left. The blot represents one out of three independent experiments with similar results. (E) Tumor progression of A498 cells injected subcutaneously in immunocompromised mice treated with ATO (n = 6) or Vehicle (n = 6) is shown on the left. Tumor volumes were measured at indicated days upon injection. The yellow box represents a period of ATO or vehicle treatment. Tumor weights at the experimental endpoint are shown on the right. In the left panels, data represent mean ± SEM (Student's t test). In the right panels, data are presented in box plots where the central band denotes the median value and boxes contain interquartile ranges. (F) Immunoblot analysis of p53 upon ATO treatment (0.25–1 μM) in the indicated cell lines. Numbers represent densitometric analysis of p53 levels normalized over vinculin. Molecular weight markers (kDa) are shown on the left. Blots represent one out of three independent experiments with similar results. (G) Proliferation assay of 786-O cells upon 5 days of ATO treatment (0.25–1 μM). Data are shown as fold change of OD 575 nm measurements over day 1. Data represent mean ± SD of three independent experiments (Student's t test). (H) Proliferation assay of 786-O cells expressing an shRNA against TP53 or a scrambled shRNA sequence (shCtrl) and treated with ATO (0.25–1 μM) for 2 or 5 days. Data are shown as fold change of OD 575 nm measurements over vehicle-treated cells and represent mean ± SD of three independent experiments (Student's t test). Source data are available online for this figure.

the formula length × diameter2 × π/6 was used to calculate tumor volumes. The maximal tumor size permitted by San Raffaele IACUC is 1 cm³ and this was not exceeded except for the in vivo experiment with A498 cells in Fig. 3, where control cells had an unexpected acceleration in tumor growth in 1 week. Therefore, only for this experiment mice where sacrificed when their tumor volumes exceeded 1 cm³. Tumors were harvested and further subjected to IHC and H&E staining.

## Lentivirus and retrovirus production

Third-generation lentivirus (LV) stocks were prepared, concentrated, and titrated as previously described (Dull et al, 1998; Follenzi et al, 2000). Briefly, self-inactivating (SIN) LV vectors were produced by transient transfection (calcium phosphate method) of HEK293T cells with the packaging plasmid pMDLg/pRRE, Rev-expressing pCMV-Rev and VSV-G envelop-encoding pMD2.VSV-G plasmids, and specific shRNA-carrying vectors. For PML knockdown experiments, Tet-pLKO-puro expressing shPML#1 (TRCN0000003867) or a scramble sequence as control were kindly donated by Arkaitz Carracedo (Martín-Martín et al, 2016). Tet-pLKO-puro expressing shPML#2 (TRCN0000003868) was created by linearizing Tet-pLKO-puro (Addgene #21915) with AgeI and EcoRI restriction enzymes and ligation of annealed shPML oligos (TRCN0000003868). Diagnostic digestions to screen for positive clones were performed using the XhoI restriction enzyme. The same procedure was applied to insert shTP53 (TRCN0000003753) in constitutive pLKO-1-puro expression vector. Retroviral particles were generated by transfecting Phoenix-AMPHO cells with pBabe-puro Empty vector (Addgene #1764) or pBabe-puro-p53DD (generated by O. Moshe) with calcium phosphate according to the de Lange laboratory protocol (https://delangelab.org/protocols).

## Gene silencing and overexpression

For lentiviral transduction, human RCC and TNBC cell lines were maintained in medium containing concentrated lentiviral supernatant and 8 μg/ml polybrene for 8 h. Cells were allowed to recover in fresh medium for 48 h before antibiotic selection. Optimized puromycin (Merck P8833) concentrations were: 1 μg/ml for RCC4, ACHN, XP258, and MDA-MB-231 cells, 0.5 μg/ml for A498, Caki-1 and 786-O cells. Experiments were conducted with bulk populations, with culture media supplemented with 10% tetracycline-free FBS (Euroclone ECS018L) to avoid promoter

leakiness of the Tet-pLKO-puro plasmids. For lentiviral transduction, RCC4 and A498 cells were maintained in medium containing retroviral particles diluted 1:1 with 8 μg/ml polybrene for 8 h. Cells were subjected to three rounds of superinfection.

## Cell proliferation, focus-forming assays, and 3T3 protocol

For cell proliferation assays, $5 \times 10^3$ RCC4, A498, XP258, 786-O, MDA-MB-231, $10 \times 10^3$ Caki-1, and $25 \times 10^3$ ACHN were plated in triplicate in 12-well plates. After 24 h of plating (day 1), one plate was fixed with PBS 4% PFA and stained with 0.1% Crystal violet solution (98 ml distilled water, 2 ml methanol, and 0,1 g Crystal Violet). In the other plates, 100 ng/ml doxycycline was used to induce shRNAs expression. Doxycycline was replenished every 48 h. Crystal violet was solubilized in 10% acetic acid, and cell proliferation was calculated by measuring crystal violet absorbance at $\lambda = 575$ nm and expressed as OD575 fold change over day 1. For focus-forming assays, cells were plated in triplicate in six-well plates ($1 \times 10^3$ cells per well) for 10 days and colony formation efficiency were evaluated with Crystal violet staining. For 3T3 protocol, $2.5 \times 10^5$ RCC4 cells were plated in p100 Petri dish (day 0). After 24 h (day 1), medium containing 100 ng ml$^{-1}$ doxycycline was used to induce the expression of shCtrl and shPML#1. Cells were allowed to grow for 72 h in a doxycycline-containing medium (day 4). Then, cells were detached, counted, and replated at $3 \times 10^5/10$ ml in doxycycline-containing medium. We repeated this procedure until day 13. We calculated the proliferation rate as total number of cells at each passage/cell plated at day 0. Data are expressed as cumulative cell number (total number at each passage x proliferation rate).

## Cell cycle analysis, BrdU incorporation, and apoptosis measurement

Cells were treated with doxycycline (100 ng/ml) for 72 h and replated at sub-confluency to permit expansion ($300 \times 10^3$ cells in p150 Petri dishes with doxycycline supplementation). The day after (96 h of doxycycline treatment), cells were pulsed with BrdU (10 μM; Merck 19–160) for 20 min to allow BrdU incorporation. Cells were fixed in PBS 70% ethanol overnight, washed two times in PBS 1% BSA and subjected to DNA denaturation using 2 N NaCl. NaCl was neutralized by adding $Na_2B_4O_7$. After washing in PBS, BrdU staining was detected with AlexaFuor 647 anti-BrdU antibody (BD Pharmingen #560209) diluted 1:100 in PBS, 1%

BSA. Hybridization occurred overnight at 4 °C. DNA was counterstained with 10 μg/ml PI (Merck P4864) supplemented with 20 μg/ml RNaseI (Merck 10109142001) to remove RNA. Samples were left at 4 °C overnight. The day after, cells were washed once in PBS and resuspended in PBS at a concentration of $100 \times 10^3/100$ μl and analyzed by FACS. For measuring apoptosis, $50 \times 10^3$ were plated in p100 Petri dishes and maintained for 96 h in doxycycline (100 ng/ml) containing media, changed every 48 h. Annexin-V staining was performed with the PE-Annexin Apoptosis Detection Kit (BD Pharmingen 559763) following the manufacturer's instructions. In all experiments, FACS analysis was performed with DB FACSCantoTM II (Becton Dickinson). Data were analyzed with FCS Express Flow Cytometry Software (De novo Software).

## Western blotting

Cell lysis was performed in RIPA buffer (Tris-HCl pH 8.0 50 mM, NaCl 150 mM, sodium deoxycholate 0.5%, sodium dodecyl sulfate 0.1%, and nonident P-40 1%) or Laemmli buffer (Bio-Rad) supplemented with protease inhibitors (Complete EDTA-free Protease Inhibitor Cocktail Tablets; Roche). After 20 min of incubation on ice, cell lysates were cleared by centrifugation (10 min at 13,500 rpm at 4 °C). Protein concentration was determined via Bradford assay. Protein extracts were diluted in 4× Laemmli buffer and boiled for 5 min at 95 °C. SDS-PAGE was performed at different polyacrylamide concentrations (7.5 or 10%) and transferred to nitrocellulose membrane through transBlot Turbo Transfer System (Bio-Rad). Membranes were blocked with 5% BSA in PBS 0.1% Tween 20 and incubated with antibodies against PML (1:1000; Novus Biologicals 100-59787), p53 clone DO-1 (1:3000 SC-126), p21 (1:500 SC-6246), LMNB1 clone B-10 (1:1000; SC-374015), β-actin (1:10,000 SC-69879), α-tubulin (1:10,000 ab4974), Vinculin (1:10,000 SC-73614). Antibodies were diluted in PBS with Tween 20 0.1% and BSA 1%. Proteins were detected with peroxidase-conjugated antibodies (Normal IgG-HRP, 1:10,000, Santa Cruz Biotechnology) using the ECL Western Blotting Detection Reagents (GE Healthcare) and bioluminescence acquired on ChemiDoc XRS+ System (Bio-Rad, Hercules, CA, USA). Densitometric analysis was performed using Fiji software.

## Immunofluorescence

For p-γH2AX and PML immunofluorescence, RCC4 and A498 cells were treated for 72 h with doxycycline (100 ng/ml) and then plated in 12-well plates on 13-mm coverslips at $20 \times 10^3$ cells/well with doxycycline. The day after (96 h of doxycycline treatment), cells were fixed 10 min at room temperature in 4% PFA and then permeabilized with 0.5% TritonX-100 in PBS. Blocking was performed in PBS, 0.05% Tween 20, 10% FBS for 30 min at room temperature and cells were incubated with antibodies against PML (PGM3) (1:1000 SC-966) and p-γH2AX Ser139 (2F3) (1:200 Biolegend 613402) diluted in PBS, 0.05% Tween 20, 1% FBS, for 1 h at room temperature. After three washes with PBS, secondary antibodies (1:500, ThermoFisher Scientific Alexafluor 488 or 546) diluted in blocking solution were used for 1 h at room temperature. Nuclei were counterstained with DAPI and coverslip mounted in Mowiol. For LAMP2, immunofluorescence cells were treated and fixed as stated above. Permeabilization was performed with 0.1% saponin in PBS, blocking with PBS 3% BSA, and coverslips were

incubated overnight at 4 °C with anti-LAMP2 antibody (1:50 ab25631) diluted in PBS 1% BSA followed by incubation with secondary antibody ThermoFisher Scientific Alexafluor 488. Images were acquired with GE HealthCare DeltaVision Ultra microscope (×60 objective) in z-stacks (stacks of 0.2 μm). Identical settings and contrasts were applied for all images of the same experiment to allow data comparison. Raw images were analyzed with Fiji software.

For PML-NBs immunofluorescence analysis upon ATO treatment and release, $20 \times 10^3$ RCC4 or MDA-MB-231 cells were treated with 1 μM ATO for 48 h, followed by ATO wash out and addition of fresh cell culture media for 3 or 6 h. Cells were fixed as stated above and permeabilized with 0.5% TritonX-100 in PBS. Anti-PML (PGM3) (1:1000 SC-966) was detected by using anti-mouse ThermoFisher Scientific Alexafluor 488 secondary antibody (1:500). Images were acquired with Zeiss Axio Observer Z.1 microscope (×63 objective) in z-stacks (stacks of 0.2 μm). Raw images were analyzed with Fiji software and PML-NBs were manually counted.

To measure PML nuclear partitioning, PML immunofluorescece images were acquired with Leica SP8 Confocal microscope. Raw images were analyzed by using CellProflier in MATLAB following the pipeline showed in Appendix Supplementary Material.

## Immunohistochemistry and hematoxylin–eosin staining

Formalin-fixed paraffin-embedded consecutive sections (4 μm) were dewaxed and hydrated through graded decrease alcohol series and stained for histology or immunohistochemical characterization. For histological analysis in bright-field microscopy, slides were stained using standard protocols for H&E (Mayer's Hematoxylin, BioOptica #05-06002/L and Eosin, BioOptica #05-10002/L). For IHC, slides were immunostained with the Automatic Leica BOND RX system (Leica Microsystems GmbH, Wetzlar, Germany). First, tissues were deparaffinized and pre-treated with the Epitope Retrieval Solution (ER1 Citrate Buffer for Ki-67 and PML) at 100 °C. Primary antibodies were developed with Bond Polymer Refine Detection (Leica, DS9800). Slides were acquired with Aperio AT2 digital scanner at a magnification of ×20 or ×10 (Leica Biosystems), and analyzed with Imagescope (Leica Biosystem). Immunohistochemical staining of KI67 was quantified using the Color Deconvolution Algorithm (Leica) following the manufacturer's instructions. For analysis, the entire slide of all samples (1 mm$^2$ each) was used.

## Transmission electron microscopy

Cultured cells were fixed as monolayers with 2.5% glutaraldehyde in 0.1 M cacodylate buffer pH 7.4 for 1 h at room temperature. After several washes in cacodylate buffer, cells were postfixed with 1% osmium tetroxide, and 1.5% potassium ferrocyanide in 0.1 M cacodylate buffer pH 7.4 for 1 h on ice. After being rinsed in dH2O, samples were stained in 0.5% uranyl acetate overnight dehydrated in increasing concentrations of ethanol, and finally embedded in Epon. Samples were cured at 60 °C in an oven for 48 h. Epon blocks were sectioned using a Leica EM UC7 ultramicrotome (Leica Microsystems). Ultrathin sections (70 nm) were collected on formvar carbon-coated slot grids, contrasted with 2% uranyl and

lead citrate. Samples were observed with a TALOS L120C Transmission Electron Microscope (ThermoFisher Scientific), and images were acquired with a CETA 4x4k CMOS camera (Thermo-Fisher Scientific).

## SA-β-gal staining

Cells were maintained in the doxycycline-supplemented medium for 72 h or passaged in the doxycycline-supplemented medium for 12 days. Then $20 \times 10^3$ cells were plated on 13 mm coverslips in 12-well plates in triplicate and maintained in doxycycline for 24 h. At 96 h or 13 days of doxycycline induction, cells were fixed with PBS 4% PFA for 10 min at room temperature. SA-β-gal staining was performed with the SA-β-gal Activity Assay Kit from Cell Signaling Technology (#23833) following the manufactures' instructions.

## RNA isolation and quantitative RT-PCR (qRT-PCR)

Total RNA from ccRCC cells was purified using RNeasy mini Kit (Quiagen). An equal amount of RNA (1 μg) was reverse transcribed through Advantage RT-for-PCR Kit (Clontech) and analyzed by qPCR using a 7900 Fast-Real Time PCR System (Applied Biosystem). The following probes for TaqMan assays were purchased from Applied Biosystem: PML HS00971694_m1, 18 S Hs03003631_g. Each sample was evaluated in technical triplicates, and data were normalized to 18 S gene. Relative expression was calculated using the comparative threshold cycle method ($2^{-\Delta\Delta Ct}$), and data are expressed as fold change in ($2^{-\Delta\Delta Ct}$) over respective shCtrl.

## RNA sequencing and data analysis

For RNA sequencing upon PML silencing, RCC4 cells expressing shCtrl or shPML#1 were treated with doxycycline (100 ng ml$^{-1}$) for 96 h. Two experiments with independent lentiviral infections were used. Each sample was processed as follows: (i) total RNA was isolated with QIAGEN RNeasy Plus Micro Kit, according to the manufacturer's instructions. (ii) RNA was treated with DNAse I (Sigma, D5307), according to the manufacturer's instructions. (iii) RNA quality was evaluated with a 2100 Bioanalyzer (Agilent) to select RNA with a RIN above 9. TruSeq stranded mRNA protocol was used for 50/30 library preparation starting from 100 ng of total RNA. Libraries were barcoded, pooled, and sequenced on an Illumina Nova-Seq 6000 sequencing system. For each run, RNA sequencing experiments were performed, generating 30 M single-end reads, 100 nucleotides long. After trimming with Trimmomatic vs 0.39, sequences were aligned using the STAR aligner (Dobin et al, 2013) to human reference genome GRCh38, and counted with feature-Counts 1.6.4 (Liao et al, 2014) on the last Gencode (Harrow et al, 2012) release for RNA sequencing. Differential gene expression was evaluated in R/BioConductor (Huber et al, 2015) using the DESeq2 package vs 1.30.1 (Love et al, 2014). A significant threshold of 0.05, adjusting the P value by FDR (False Discovery Rate) was established to identify differentially expressed genes. A log2 Fold Change (log2FC) cut-off of 0.5 was used to identify up- (log2FC > 0.5) and down- (log2FC < −0.5) regulated genes. Functional enrichment analyses were performed using EnrichR (Kuleshov et al, 2016). Reactome, KEGG, MSigDB_Hallmark, WikiPathway, and Gene Ontologies, were used as reference databases,

### The paper explained

#### Problem

Clear-cell renal cell carcinoma (ccRCC) poses a significant challenge in cancer treatment, with high rates of resistance to radio and chemotherapy and poor patients' outcomes at late diagnosis. The *TP53* gene is rarely mutated in ccRCC, but the wild-type p53 protein is kept in a functionally inactive state that renders this disease refractory to many available treatments. Therefore, strategies aimed to reactivate the p53 pathway represent a valuable avenue to develop new therapeutic approaches for ccRCC.

#### Results

Here we identified a novel ccRCC dependency to the *PML* gene, whose inhibition hinders ccRCC expansion. Importantly, we found that genetic inactivation of PML induces cellular senescence and growth arrest by releasing the tumor-suppressive function of p53. Moreover, pharmacological targeting of PML with arsenic trioxide, an FDA-approved drug used as first-line therapy in leukemia, reactivates p53 and delays ccRCC tumor growth.

#### Impact

Our findings shed light on a previously unrecognized dependency of ccRCC on *PML* expression and reveal a novel regulatory mechanism of p53 inactivation in this tumor type. Pharmacological inhibition of PML with an FDA-approved drug impairs tumor expansion in preclinical ccRCC models. Thus, our studies suggest that arsenic trioxide may be repurposed as a ccRCC anticancer drug and pave the way for further investigation into the clinical utility of targeting PML to improve ccRCC outcome.

and significant pathways were filtered by adjusted $P$ value < 0.1. ggplot2 in R (https://ggplot2.tidyverse.org/) was used for plots.

## Analysis of PML expression in publicly available datasets

Proteomic (CPTAC-KIRC, CPTAC-GBM, CPTAC-BRCA) and transcriptomic (TCGA-KIRC, TCGA-KIRP, TCGA-GBM, TCGA-HNSC) datasets were interrogated for PML and UBC9 expression and patients' survival probability using the UALCAN web tool (Chandrashekar et al, 2017) (online source: http://ualcan.path.uab.edu). *PML* and *UBE2I* mRNA abundance in cancer cell lines was queried in the Cancer Cell Line Encyclopedia dataset using cBioPortal (Cerami et al, 2012; Gao et al, 2013) (online source: https://www.cbioportal.org).

## Statistical analysis

Animals used for in vivo experiments with arsenic trioxide were randomized into two treatment groups (vehicle or arsenic trioxide) such that tumor volume was similar between the groups. The experiments were conducted as non-blind tests and no mice were excluded from the experiments. Unless otherwise specified in the figure legends, all experiments reported in this study were performed using at least three independent experiments or biological replicates. Analysis of statistical significance between two groups was determined by the two-tailed Student's *t* test, and a level of confidence of 0.05 was accepted for statistical significance. All analyses were performed using GraphPad Prism version 8.2.1

for macOS (GraphPad Software, San Diego, CA, USA, www.graphpad.com).

## Data availability

Data underlying this study have been deposited in Gene Expression Omnibus (GEO) at GSE246846.

The source data of this paper are collected in the following database record: biostudies:S-SCDT-10_1038-S44321-024-00077-3.

## Peer review information

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

## Acknowledgements

The authors would like to thank Dr. Arkaitz Carracedo for kindly providing inducible shRNA constructs (CICbioGUNE, Bilbao, Spain) and Prof. Dr. Ian Frew (Freiburg University, Germany), Dr. Simone Cardaci (IRCCS San Raffaele Scientific Institute, Italy) and all members of the Bernardi laboratory for valuable discussion and support. All authors participated in the discussion of the results and provided useful comments and suggestions for the realization of the manuscript. The authors would like to thank the following San Raffaele Hospital Core Facilities: Flow Cytometry Resource, Advanced Cytometry Technical Applications Laboratory (FRACTAL) for help in performing FACS experiments; Advanced Light and Electron Microscopy BioImaging Center (ALEMBIC) for imaging studies; Animal Histopathology Facility for H&E and immunohistochemistry on ccRCC xenograft tumors. This work was supported by the Italian Association for Cancer Research (AIRC) with an Investigator Grant grant to RB (ID: 20170).

## Author contributions

**Matilde Simoni**: Conceptualization; Data curation; Formal analysis; Validation; Investigation; Visualization; Methodology; Writing—original draft; Writing—review and editing. **Chiara Menegazzi**: Investigation; Methodology. **Cristina Fracassi**: Visualization; Methodology. **Claudia C Biffi**: Investigation; Methodology. **Francesca Genova**: Data curation; Formal analysis; Methodology. **Nazario Pio Tenace**: Methodology. **Roberta Lucianò**: Formal analysis; Investigation; Visualization; Methodology. **Andrea Raimondi**: Investigation; Visualization; Methodology. **Carlo Tacchetti**: Investigation; Visualization. **James Brugarolas**: Conceptualization; Resources.

**Davide Mazza**: Resources; Data curation; Formal analysis; Validation; Methodology. **Rosa Bernardi**: Conceptualization; Supervision; Funding acquisition; Writing—original draft; Project administration; Writing—review and editing.

Source data underlying figure panels in this paper may have individual authorship assigned. Where available, figure panel/source data authorship is listed in the following database record: biostudies:S-SCDT-10_1038-S44321-024-00077-3.

## Disclosure and competing interests statement

The authors declare no competing interests.

# Expanded View Figures

**Figure EV1.   High *PML* mRNA expression correlates with decreased ccRCC survival.**

(**A–D**) Kaplan–Meier curves showing the survival probability of patients with high or low PML mRNA in the indicated TCGA datasets: KIRC ($n = 72$ normal and $n = 533$ tumor samples); KIRP ($n = 32$ normal and $n = 290$ tumor samples) GBM ($n = 5$ normal and $n = 156$ tumor samples); HNSC ($n = 44$ normal and $n = 520$ tumor samples). *P* values were obtained by log rank test. (Source: UALCAN). (**E**) Representative images of PML immunofluorescence (green) in the indicated ccRCC and TNBC cell lines. Scale bar 10 μm. Nuclei were counterstained with DAPI (blue). Quantification data showing the distribution of PML-NBs/cell in each cell line ($n = 30$). (**F–I**) Box and whisker plots with individual data points showing the distribution of nuclear area (**E**), PML nuclear intensity (**F**), number of PML-NBs/nucleus (**G**) and number of PML-NBs per unit area (**H**) in RCC4 and MDA-MB-231 cells (RCC4, $n = 124$; MDA-MB-231, $n = 150$). The central band denotes the median value, box contains interquartile ranges, while whiskers mark minimum and maximum values. Statistical significance was calculated with Bonferroni corrected Kolmogorov–Smirnov test.

**A**

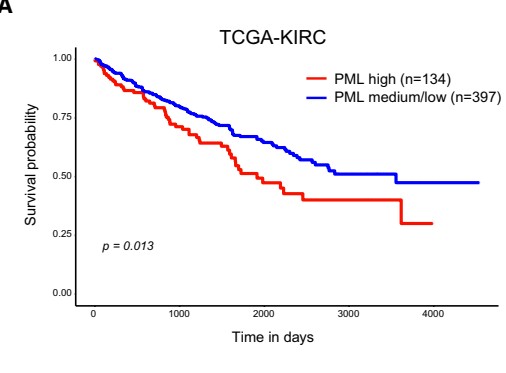

**B**

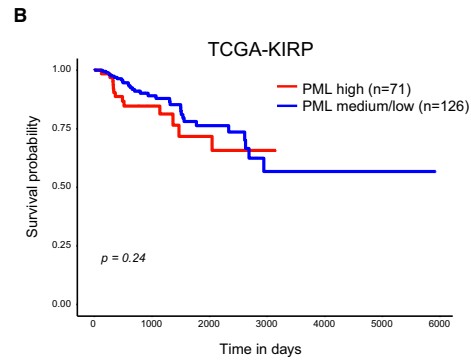

**C**

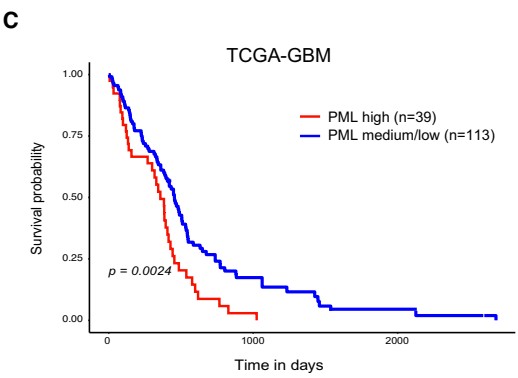

**D**

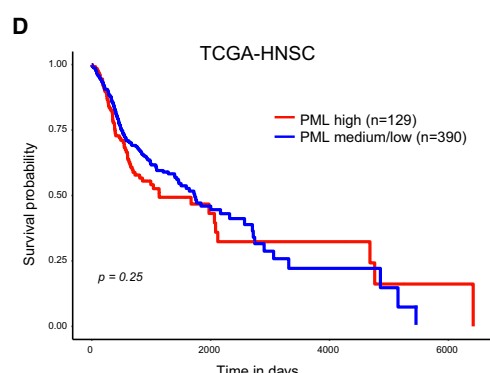

**E**

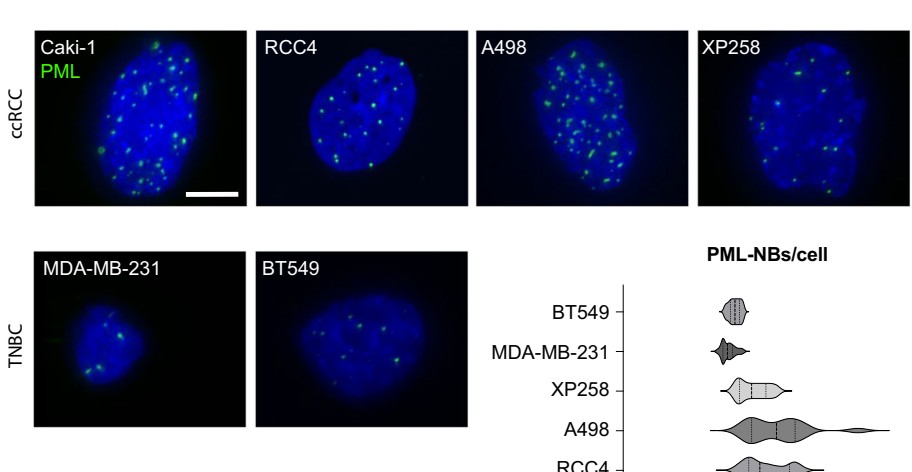

**F**

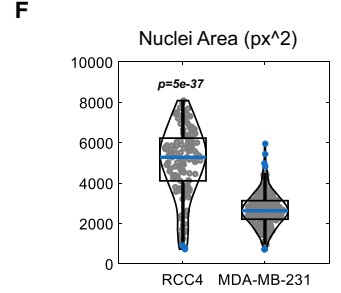

**G**

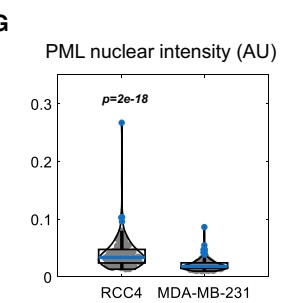

**H**

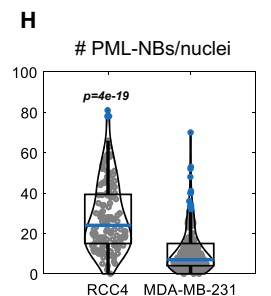

**I**

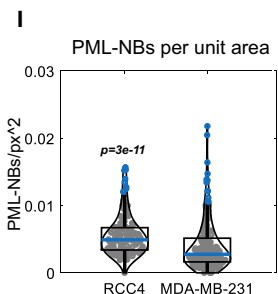

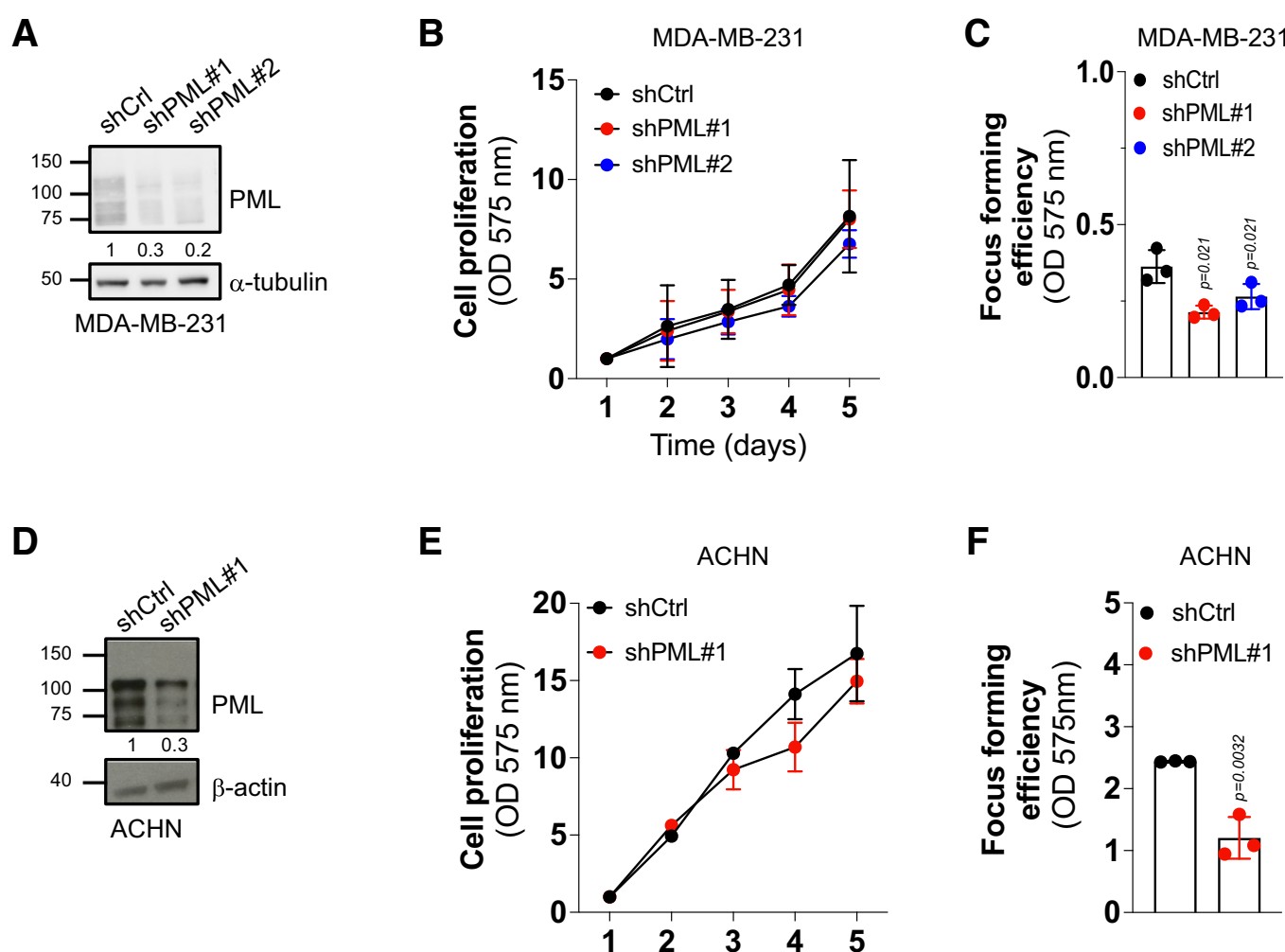

**Figure EV2. PML silencing in TNBC and papillary RCC cell lines affects focus forming efficiency.**

(A) Immunoblot analysis showing silencing efficiency of two independent shRNAs against PML (shPML#1 and shPML#2) compared to a scrambled shRNA sequence (shCtrl) in MDA-MB-231 cells. β-actin was used as loading control. Numbers represent densitometric analysis of PML levels normalized over α-tubulin. Molecular weight markers (kDa) are shown on the left. The blot represents one out of three independent experiments with similar results. (B, C) Proliferation (B) and focus-forming (C) assays performed in MDA-MB-231 cell line expressing shPML#1, shPML#2, or shCtrl. Data represent mean ± SD of three biological replicates (Student's *t* test). (D) Immunoblot analysis showing silencing efficiency of shPML#1 compared to shCtrl in ACHN cell line. β-actin was used as loading control. Numbers represent densitometric analysis of PML levels normalized over α-tubulin. Molecular weight markers (kDa) are shown on the left. The blot represents one out of three independent experiments with similar results. (E, F) Proliferation (E) and focus-forming (F) assays performed in ACHN cell line expressing shPML#1 or shCtrl. Data represent mean ± SD of three biological replicates (Student's *t* test).

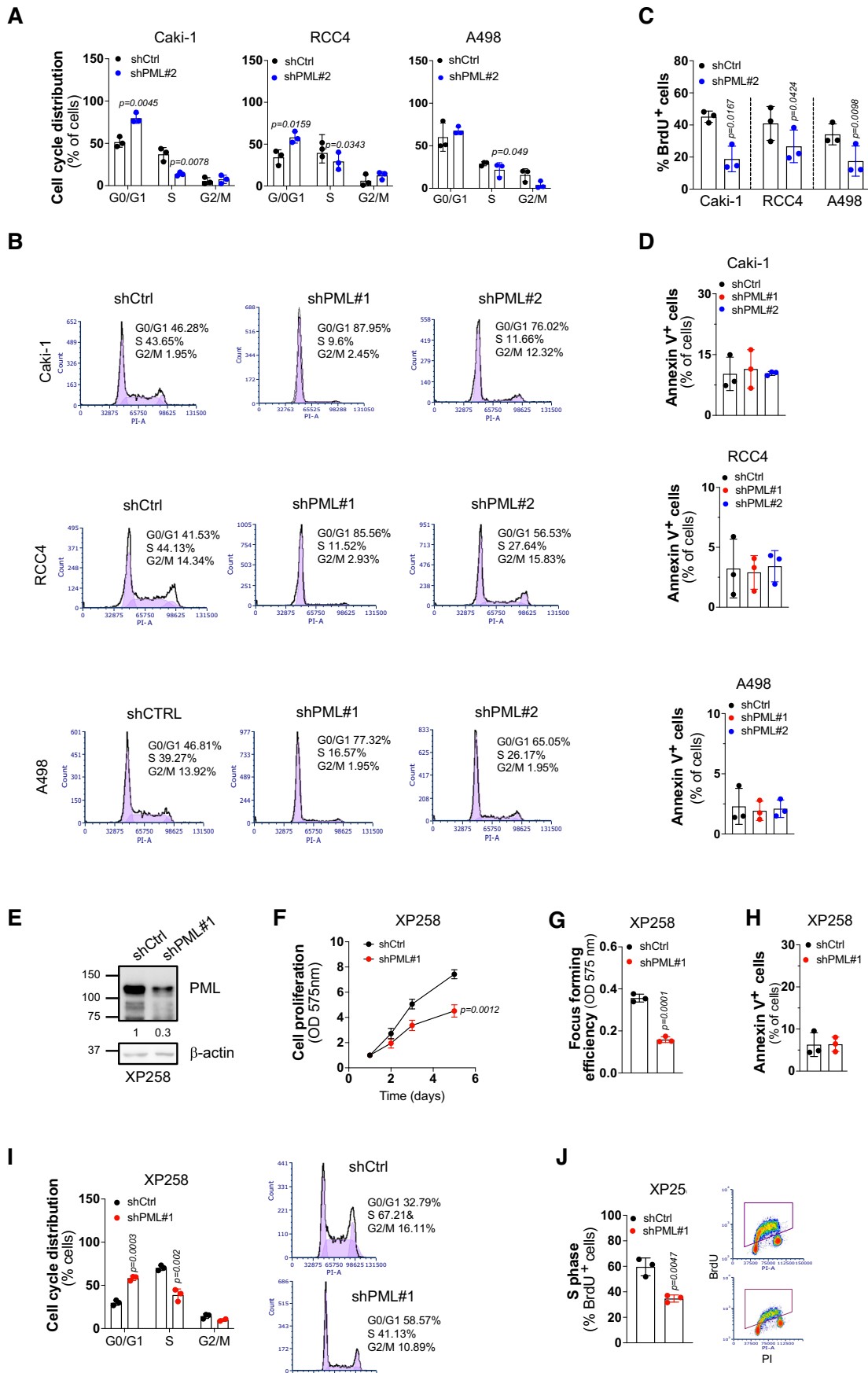

◀ **Figure EV3. PML silencing inhibits ccRCC proliferation and cell cycle progression.**

(A) Cell cycle distribution by FACS in the indicated ccRCC cell lines expressing shPML#2, or shCtrl. Data represent mean ± SD of three biological replicates (Student's *t* test). (B) Representative FACS profiles of the indicated cell lines expressing shPML#1, shPML#2 or shCtrl. Shown are the results of one out of three independent experiments with similar results. (C) Percentage of BrdU-positive cells in the indicated ccRCC cell lines expressing shPML#2 or shCtrl. Data represent mean ± SD of three biological replicates (Student's *t* test). (D) Percentage of Annexin-V positive cells in the indicated cell lines expressing shPML#1, shPML#2 or shCtrl. Data represent mean ± SD of three biological replicates (Student's *t* test). (E) Immunoblot analysis showing silencing efficiency of shPML#1 in XP258 PDX-derived cell line. β-actin was used as loading control. Numbers represent densitometric analysis of PML levels normalized over α-tubulin. Molecular weight markers (kDa) are shown on the left. The blot represents one out of three independent experiments with similar results. (F, G) Proliferation (F) and focus-forming (G) assays performed in XP258 PDX-derived cell line with shPML#1 or shCtrl. Data represent mean ± SD of three biological replicates (Student's *t* test). (H) Percentage of Annexin-V positive cells in XP250 PDX-derived cell line expressing shPML#1 or shCtrl. Data represent mean ± SD of three biological replicates (Student's *t* test). (I) Cell cycle distribution by FACS of XP258 PDX-derived cell line expressing shPML#1 or shCtrl (left). Data represent mean ± SD of three biological replicates (Student's *t* test). Representative FACS profiles of cell cycle distribution of XP258 PDX-derived cell line expressing shPML#1 or shCtrl (right). Shown are the results of one out of three independent experiments with similar results. (J) Percentage of BrdU-positive cells in XP258 cell line expressing shCtrl or shPML#1 (left). Representative scatter plot of BrdU postitive cells by FACS analysis (right). Shown are the results of one out of three independent experiments with similar results. In the left panel, data represent mean ± SD of three biological replicates (Student's *t* test).

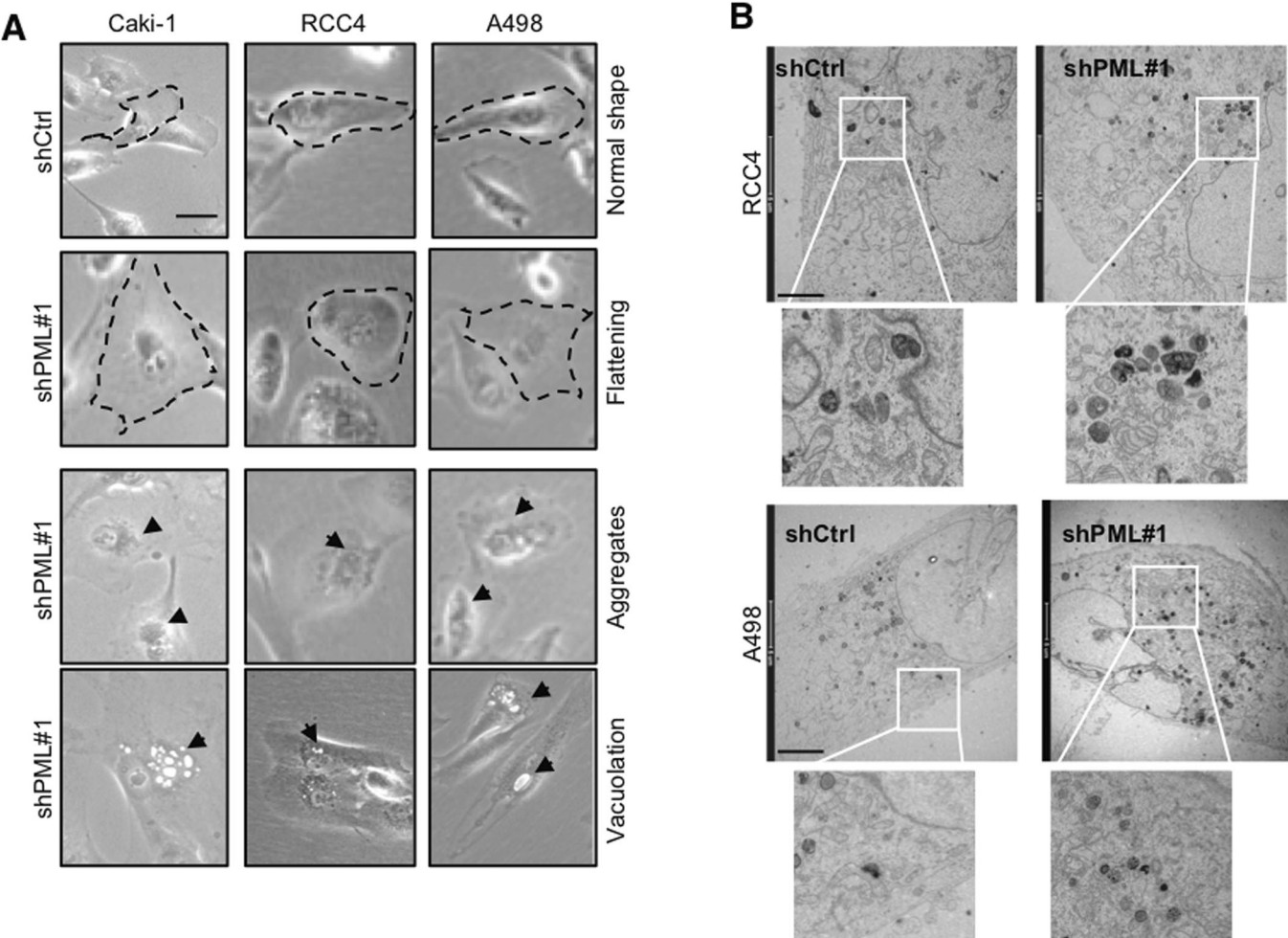

**Figure EV4.  PML knockdown recapitulates morphological features of cellular senescence.**

(A) Representative phase contrast images of the indicated cell lines expressing shPML#1 or shCtrl. Black arrowheads indicate cytoplasmic aggregates and vacuoles. Scale bar 20 μm. Shown are the results of one out of three experiments with similar results. (B) Representative transmission electron microscopy of RCC4 and A498 cell lines expressing shPML#1 or shCtrl. Indents show higher magnifications of degradative structures in boxed areas. Scale bar 5 μm. Shown are the results of one out of two independent experiments with similar results.

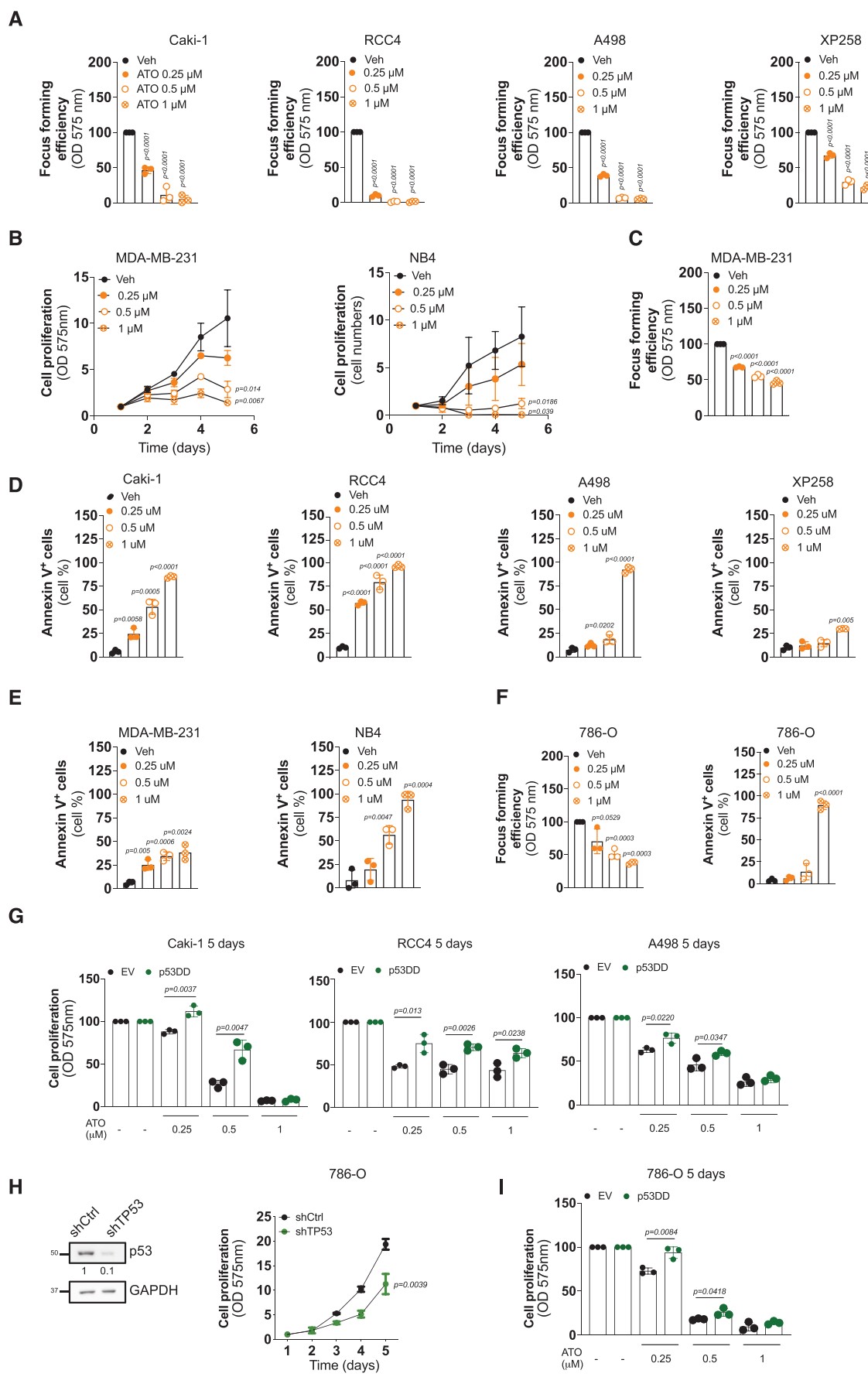

◀ **Figure EV5. ATO inhibits cell proliferation and induces apoptosis in ccRCC, TNBC and APL cells.**

(A) Focus-forming efficiency of the indicated cell lines upon 14 days of ATO treatment (0.25–1 μM). Data represent mean ± SD of three independent experiments (Student's *t* test). (B, C) Cell proliferation of MDA-MB-231 and NB4 cells (B) and focus-forming assay of MDA-MB-231 cells (C) upon 5 days (B) and 14 days (C) of ATO treatment (0.25–1 μM). For MDA-MB-231 cells, data are shown as fold change of OD 575 nm measurements over day 1. For NB4 cells, data are shown as normalized cell numbers over day 1. Data represent mean ± SD of three independent experiments (Student's *t* test). (D, E) Percentage of Annexin-V positive cells of the indicated cell lines upon 5 days of ATO treatment (0.25–1 μM). Data represent mean ± SD of three independent experiments (Student's *t* test). (F) Focus-forming assay of 786-O cells upon 14 days of ATO (0.25–1 μM) treatment is shown on the left. Percentage of 786-O Annexin-V positive cells upon 5 days of ATO (0.25–1 μM) treatment is shown on the right. Data represent mean ± SD of three independent experiments (Student's *t* test). (G) Cell proliferation of the indicated cell lines upon 5 days of treatment with ATO (0.25–1 μM). Data are represented as normalized values of OD 575 nm measurements over vechicle-treated cells. Data represent mean ± SD of three independent experiments (Student's *t* test). (H) Immunoblot analysis showing the silencing efficiency of TP53 shRNA (shTP53) compared to a scramble sequence (shCtrl) in 786-O cells (left). GAPDH was used as loading control. Numbers represent densitometric analysis of p53 levels normalized over GAPDH. The blot represents one out of three independent experiments with similar results. Proliferation assay of 786-O cells expressing shCtrl or shTP53 (right). Data are shown as fold change of OD 575 nm measurements over day 1. Data represent mean ± SD of three independent experiments (Student's *t* test). (I) Cell proliferation of 786-O cells upon 5 days of treatment with ATO (0.25–1 μM). Data are represented as normalized values of OD 575 nm measurements over vechicle-treated cells. Data represent mean ± SD of three independent experiments (Student's *t* test).

