## [Peer Review File · EMBO Molecular Medicine]

PML restrains p53 activity and cellular senescence in clear cell renal cell carcinoma

Matilde Simoni, Chiara Menegazzi, Cristina Fracassi, Claudia Biffi, Francesca Genova, Nazario Pio Tenace, Roberta Lucianò, Andrea Raimondi, Carlo Tacchetti, James Brugarolas, Davide Mazza, and Rosa Bernardi

Corresponding author: Rosa Bernardi (bernardi.rosa@hsr.it)

Review Timeline:

Submission Date:	22nd Feb 24
Editorial Decision:	13th Mar 24
Revision Received:	16th Apr 24
Editorial Decision:	22nd Apr 24
Revision Received:	23rd Apr 24
Accepted:	25th Apr 24

Editor: Lise Roth

Transaction Report:

(Note: Please note that the manuscript was transferred from another journal where it was originally reviewed. Since the original reviews are not subject to EMBO's transparent review process policy, the reports and author response cannot be published. With the exception of the correction of typographical or spelling errors that could be a source of ambiguity, letters and reports are not edited. Depending on transfer agreements, referee reports obtained elsewhere may or may not be included in this compilation. Referee reports are anonymous unless the Referee chooses to sign their reports.)

13th Mar 2024

Dear Dr. Bernardi,

Thank you for the submission of your manuscript to EMBO Molecular Medicine. I have sent it to a single reviewer for evaluation, together with the existing referees' reports from the other journal, and your point-by-point rebuttal letter. We have now received the enclosed report from this reviewer. As you will see, he/she is positive and supportive of publication pending revisions of the manuscript according to your rebuttal letter.

In particular, it will be important to characterize PML NB formation in the different cell lines used, to show whether ccRCC have SUMOylation defects, and to show the effects of arsenic in P53-mutant 786-O cells. Please also address the minor comments from our reviewer.

EMBO Molecular Medicine encourages a single round of revision only and therefore, acceptance or rejection of the manuscript will depend on the completeness of your responses included in the next, final version of the manuscript. For this reason, and to save you from any frustrations in the end, I would strongly advise against returning an incomplete revision.

We are expecting your revised manuscript within three months, if you anticipate any delay, please contact us.

We require:

4) A .docx formatted letter INCLUDING the reviewers' reports and your detailed point-by-point responses to their comments. As part of the EMBO Press transparent editorial process, the point-by-point response is part of the Review Process File (RPF), which will be published alongside your paper.

5) A complete author checklist, which you can download from our author guidelines (<https://www.embopress.org/page/journal/17574684/authorguide#submissionofrevisions>). Please insert information in the checklist that is also reflected in the manuscript. The completed author checklist will also be part of the RPF.

6) Please note that all corresponding authors are required to supply an ORCID ID for their name upon submission of a revised manuscript.

7) It is mandatory to include a 'Data Availability' section after the Materials and Methods. Before submitting your revision, primary datasets produced in this study need to be deposited in an appropriate public database, and the accession numbers and database listed under 'Data Availability'. Please remember to provide a reviewer password if the datasets are not yet public (see <https://www.embopress.org/page/journal/17574684/authorguide#dataavailability>).

In case you have no data that requires deposition in a public database, please state so in this section ("This study includes no data deposited in external repositories.").

Note that the Data Availability Section is restricted to new primary data that are part of this study.

8) For data quantification: please specify the name of the statistical test used to generate error bars and P values, the number (n) of independent experiments (specify technical or biological replicates) underlying each data point and the test used to calculate p-values in each figure legend. The figure legends should contain a basic description of n, P and the test applied. Graphs must include a description of the bars and the error bars (s.d., s.e.m.). Please provide exact p values.

9) Our journal encourages inclusion of *data citations in the reference list* to directly cite datasets that were re-used and

obtained from public databases. Data citations in the article text are distinct from normal bibliographical citations and should directly link to the database records from which the data can be accessed. In the main text, data citations are formatted as follows: "Data ref: Smith et al, 2001" or "Data ref: NCBI Sequence Read Archive PRJNA342805, 2017". In the Reference list, data citations must be labeled with "[DATASET]". A data reference must provide the database name, accession number/identifiers and a resolvable link to the landing page from which the data can be accessed at the end of the reference. Further instructions are available at .

13) Author contributions: CRediT has replaced the traditional author contributions section because it offers a systematic machine readable author contributions format that allows for more effective research assessment. Please remove the Authors Contributions from the manuscript and use the free text boxes beneath each contributing author's name in our system to add specific details on the author's contribution. More information is available in our guide to authors.

16) As part of the EMBO Publications transparent editorial process initiative (see our Editorial at <http://embomolmed.embopress.org/content/2/9/329>), EMBO Molecular Medicine will publish online a Review Process File (RPF) to accompany accepted manuscripts.

In the event of acceptance, this file will be published in conjunction with your paper and will include the anonymous referee reports, your point-by-point response and all pertinent correspondence relating to the manuscript. Let us know whether you agree with the publication of the RPF and as here, if you want to remove or not any figures from it prior to publication. Please note that the Authors checklist will be published at the end of the RPF.

I look forward to receiving your revised manuscript.

With kind regards,

Lise Roth

***** Reviewer's comments *****

Referee #1 (Remarks for Author):

I have reviewed the manuscript by Simoni et al., including their point-by-point response to two sets of referee comments. The authors show that PML is upregulated in clear cell renal cell carcinoma (ccRCC) and that PML inhibition by RNAi reduces ccRCC proliferation and tumorigenesis. They link PML inhibition to senescence via p53 activation and show that ATO, which is known to inhibit PML, also suppresses ccRCC formation in xenografts. The study is presented clearly and the data look convincing. The new results are discussed in the context of previous results and the novelty of the results is highlighted throughout the text.

The study has previously been reviewed by two experts who in my opinion provide a critical but fair evaluation of the work. They suggest a number of ways in which the study could be improved, and the authors present well-articulated responses to these comments, suggesting specific experiments that would address the criticism and providing necessary clarifications.

In my opinion the authors' plan for revisions is sufficient and I would like to make only a couple of additional comments.

1. The authors use multiple cell lines to demonstrate the importance of PML for the proliferation of renal carcinoma cells, which looks convincing. In genome wide CRISPR / Cas9 screens in the Cancer DepMap project (<https://depmap.org/>), PML inhibition has not, however, shown a clear proliferative phenotype in ccRCC cell lines. It would be interesting to include the authors' thoughts of these data in the manuscript. Could the discrepancy be somehow addressed?
2. Related to the proposed role of TP53 mutations as determinants of ATO or PML inhibition sensitivity, while 786-O cells carry a TP53 mutation, based on DepMap and other screening data TP53 inhibition still provides 786-O cells with a detectable proliferative advantage, suggesting that they may not be completely null for TP53 activity. Using a different TP53 mutant ccRCC cell line could thus be helpful. Alternatively, TP53 could be mutated in PML inhibition sensitive cells.

Overall, this is a very interesting study that provides novel insight into PML function and renal carcinogenesis.

Response to referees #1 and #2 from previous journal removed.

Referee #3

I have reviewed the manuscript by Simoni et al., including their point-by-point response to two sets of referee comments. The authors show that PML is upregulated in clear cell renal cell carcinoma (ccRCC) and that PML inhibition by RNAi reduces ccRCC proliferation and tumorigenesis. They link PML inhibition to senescence via p53 activation and show that ATO, which is known to inhibit PML, also suppresses ccRCC formation in xenografts. The study is presented clearly and the data look convincing. The new results are discussed in the context of previous results and the novelty of the results is highlighted throughout the text.

The study has previously been reviewed by two experts who in my opinion provide a critical but fair evaluation of the work. They suggest a number of ways in which the study could be improved, and the authors present well-articulated responses to these comments, suggesting specific experiments that would address the criticism and providing necessary clarifications.

In my opinion the authors' plan for revisions is sufficient and I would like to make only a couple of additional comments.

We thank this referee for her/his kind comments.

1. The authors use multiple cell lines to demonstrate the importance of PML for the proliferation of renal carcinoma cells, which looks convincing. In genome wide CRISPR/Cas9 screens in the Cancer DepMap project (<https://depmap.org/>), PML inhibition has not, however, shown a clear proliferative phenotype in ccRCC cell lines. It would be interesting to include the authors' thoughts of these data in the manuscript. Could the discrepancy be somehow addressed?

We thank the referee for this comment. We agree about the apparent discrepancy of these results and we do not have a conclusive answer as to why this is the case. We have given the question profound thinking but because we are not fully familiar with the wet and bioinformatics methodologies that are used by DepMap, we can only come up with some speculations. Specifically, we could not find any information about controlling the efficiency of knock-out or knock-down of single genes (i.e., PML) upon transduction of genome wide sgRNA or shRNA libraries at the cell population level. Thus, we cannot exclude that PML is inefficiently downregulated by RNAis or sgRNAs in DepMap screenings. Along these lines, it is unclear to us what is the efficacy of full knock-out with respect to heterozygous deletion by sgRNAs used in the DepMap project, and whether this has been calculated at a global or single gene level. Finally, and most importantly, it is our understanding that efforts like DepMap were developed to assay gene dependencies at a population level based on loss of cell viability. Because PML silencing induces a state of prolonged cell cycle arrest, we cannot predict if and how the maintenance of live, cell cycle-arrested cells with low PML expression may impact on calculations that were designed to identify cells depleted by cell death in the general population. However, we agree that this apparent discrepancy should be discussed, and we have now added a statement in the Discussion section.

2. Related to the proposed role of TP53 mutations as determinants of ATO or PML inhibition sensitivity, while 786-O cells carry a TP53 mutation, based on DepMap and other screening data TP53 inhibition still provides 786-O cells with a detectable proliferative advantage, suggesting that they may not be completely null for TP53 activity. Using a different TP53 mutant ccRCC cell line could thus be helpful. Alternatively, TP53 could be mutated in PML inhibition sensitive cells.

We thank this referee for her/his important comment. The referee rightly observes that TP53 inhibition in 786-O cells leads to detectable proliferative advantage in DepMap, indicating that these cells are not completely null for TP53 activity. We have obtained similar results by using a specific TP53 shRNA, as we now report in Fig EV5H. These observations suggest that 786-O cells may contain a p53 gain-of-function mutant. Although different types of TP53 mutations are reported in

the literature in 786-O cells perhaps obtained from different sources, our cells were obtained from the ATCC and carry two TP53 mutations: c.560-2A>G and c.832C>G. c.560-2A>G occurs in a splicing acceptor site and is reportedly a mutation of unknown significance (Leroy et al., *Human Mutation*, 2014). c.832C>G causes a P278A substitution that is also poorly characterized, as stated by Chen et al., *Cancer Cell*, 2021, who studied the reactivation of TP53 mutants by ATO. Interestingly, although 786-O cells were not analyzed by Chen et al., we found that they contain at least a TP53 mutant that can be reactivated by ATO. We show this by treating 786-O cells with ATO and inhibiting endogenous TP53 expression/activity via two approaches (shRNA-mediated TP53 silencing and TP53 inhibition with a dominant negative mutant; now shown in Fig 7H and Fig EV5I). Taken together, these experiments suggest that 786-O cells carry a gain-of-function TP53 mutant and at least one TP53 mutant that can be reactivated by ATO, as we now discuss in the Results section.

Regarding the suggestion to extend our observations to an additional ccRCC cell line, we do not have access to other TP53 mutant ccRCC cell lines, but Chen et al. showed that within the few TP53 mutant ccRCC cell lines, RFX393 cells carry a R175H mutation, which is a typical ATO-reactivated mutant, thus pointing to an additional cell line that is sensitive to ATO-mediated reactivation.

Overall, this is a very interesting study that provides novel insight into PML function and renal carcinogenesis.

22nd Apr 2024

Dear Dr. Bernardi,

Thank you for submitting your revised manuscript. We have now received the report from the referee re-reviewed your manuscript, and as you will see below, this referee is satisfied with the revisions. I will therefore be able to accept your manuscript once the following minor editorial points are addressed:

1/ Manuscript text:

- Our notification email for James Brugarolas bounced; please provide a valid email address.
- Please remove the red text, and only keep in track changes mode any new modification.
- Please provide up to 5 keywords.
- Materials and Methods:
 - o Cells: please indicate whether the cells were authenticated.
 - o Mice: please indicate housing and husbandry conditions.
 - o Statistics: please include a statement on randomization, blinding, and inclusion/exclusion criteria.
- Data availability: Thank you for providing a reviewer token. Please note that data must be public before acceptance of the manuscript.

2/ Figures:

- Table EV1 should be renamed Dataset EV1
- Please address the queries from our data editors in the figure legends:
 1. Please note that the legend for figures EV 1f-i is incorrectly labelled as EV 1e-h. This needs to be rectified.
 2. Please note that the legend for figure EV 5i is missing in the manuscript. This needs to be rectified.
 3. Please indicate the statistical test used for data analysis in the legends of supplementary figures 1b, d; EV 1a-d.
 4. Please note that the box plots need to be defined in terms of minima, maxima, centre, bounds of box and whiskers, and percentile in the legends of figures 1b, d-e, h-i; EV 1f-i.
 5. Please note that scale bar and its definition are missing for figure EV 4b.
 6. Please note that the black arrows are not defined in the legend of figure EV 4a. This needs to be rectified.

3/ Thank you for providing Source Data.

- please upload them as one file per figure.
- please carefully check the labeling, in particular Figure 2D, excel file.
- Source Data for figure 7F are missing.

4/ Checklist:

- Please fill in authors and manuscript information on the top left corner.
- Please complete the "Experimental study design and statistics" section.

5/ Thank you for providing a synopsis text and image. Could you provide a higher resolution image, with a white background?

6/ As part of the EMBO Publications transparent editorial process initiative (see our Editorial at <http://embomolmed.embopress.org/content/2/9/329>), EMBO Molecular Medicine will publish online a Review Process File (RPF) to accompany accepted manuscripts.

This file will be published in conjunction with your paper and will include the anonymous referee reports, your point-by-point response and all pertinent correspondence relating to the manuscript. This file will not include previous review process. Let us know whether you agree with the publication of the RPF and as here, if you want to remove or not any figures from it prior to publication.

I look forward to receiving your revised manuscript at your earliest convenience.

With kind regards,

Lise

**** Reviewer's comments ****

Referee #1 (Remarks for Author):

The authors have addressed the previously raised points satisfactorily. I have no further comments.

The authors addressed the minor editorial issues.

25th Apr 2024

Dear Dr. Bernardi,

Thank you for submitting your revised files. I am pleased to inform you that your manuscript is accepted for publication and is now being sent to our publisher to be included in the next available issue of EMBO Molecular Medicine!

If you have any questions, please do not hesitate to contact the Editorial Office.

Congratulations on your interesting work!

With kind regards,

Lise
